# A Modified Functional Observer-Based EID Estimator for Unknown Continuous-Time Singular Systems

**Jason Sheng-Hong Tsai [1], Yang-Fang Chen [1], Te-Jen Su [2,3,\*], Chia-Yuan Chang [1], Shu-Mei Guo [4], Leang-San Shieh [5] and Jose I. Canelon [6]**

[1] Department of Electrical Engineering, National Cheng-Kung University, Tainan 701, Taiwan; shtsai@mail.ncku.edu.tw (J.S.-H.T.); yfchen821201@gmail.com (Y.-F.C.); yuan010134@gmail.com (C.-Y.C.)

[2] Department of Electronic Engineering, National Kaohsiung University of Science and Technology, Kaohsiung 800–852, Taiwan

[3] Graduate Institute of Clinical Medicine, Kaohsiung Medical University, Kaohsiung 800–852, Taiwan

[4] Department of Computer Science and Information Engineering, National Cheng-Kung University, Tainan 701, Taiwan; guosm@mail.ncku.edu.tw

[5] Department of Electrical and Computer Engineering, University of Houston, Houston, TX 77204-4005, USA; lshieh@central.uh.edu

[6] Electrical Engineering School, Universidad del Zulia, Maracaibo 4005, Venezuela; joseicanelon@gmail.com

\* Correspondence: sutj@nkust.edu.tw

**Abstract:** This paper presents the design of a linear quadratic analog tracker (LQAT) based on the observer–Kalman-filter identification (OKID) method and the design of a modified functional observer-based equivalent input disturbance (EID) estimator for unknown square–non-square singular analog systems with unknown input and output disturbances. First, an equivalent mathematical model of the singular analog system is presented to simulate the time response of continuous-time linear singular analog systems to arbitrary inputs via the model conversion method. Then, for the unknown singular analog system, it constructs a linear quadratic analog tracker with state feedback and feed-forward gains based on the off-line OKID method. Furthermore, it extends the design methodology of the EID estimator for strictly proper regular systems with unknown matched–mismatched input and output disturbances to proper regular systems. It is important to mention that the newly developed modified functional observer for proper systems is used to estimate the unknown EID of singular analog systems and that the constraints on the dimensions of unknown disturbances can be eliminated by using the newly proposed EID estimation method. The contributions of this paper can be listed as follows: (1) based on both the OKID method and the discrete-to-continuous model conversion, the simulation of the time responses of the continuous-time linear singular models (which are not feasible using existing MATLAB toolboxes) become feasible; (2) for effective control of the unknown singular analog system, an off-line OKID method is proposed to design an LQAT with state feedback and feed-forward gains; and (3) based on the newly developed modified functional observer for the reduced-order proper regular system, the original EID estimator in the literature is newly extended to estimate the EID from the unknown strictly proper singular analog system, without the original dimensional constraints of the disturbances. It is important to mention that the disturbances of interest can be unknown matched–mismatched input and output disturbances.

**Keywords:** singular systems; generalized Riccati equation; input–output direct feed-through term; functional observer; observer–Kalman-filter identification; equivalent input disturbance (EID); disturbance estimator

## 1. Introduction

Singular analog system models naturally arise in describing large-scale (complex) systems, such as interconnected power systems. In general, a large class of interconnected state-variable subsystems is described as a singular system, even though the entitled state–space representation may not be available. Practical singular systems usually comprise dynamic and non-dynamic subsystems, which are mathematically represented by a combination of differential and algebraic equations. The complex features of singular systems (also known as descriptor systems) often make it difficult to find numerical and analytical solutions for such systems, especially for control tasks.

The numerically stable and computationally fast matrix sign function-based method has been used to solve the regulator-based generalized algebraic Riccati equation for optimal control of a linear continuous-time singular system [1]. A technique was developed in [2] to decompose the singular system into a reduced-order regular subsystem and a non-dynamic subsystem. Then, an optimal tracker was developed in [3] based on the equivalent model of the linear singular system. Nevertheless, an approach for solving the tracker-based generalized algebraic Riccati equation for the singular system has not yet been developed in the literature. It was proven in [4] that the tracker-based generalized algebraic Riccati equation cannot be directly solved for a singular system, even with an impulsive model-free and strictly proper transfer function. To overcome this issue, in [4] the singular system is converted into an equivalent regular model with a direct transmission term from input to output. Based on this equivalent proper model, the approach for solving the regulator-based generalized algebraic Riccati equation can then be extended to solve the tracker-based generalized algebraic Riccati equation. In other words, the optimal control methodology for finding the linear quadratic regulator via the matrix sign function [1] can be directly extended to find the optimal tracker for singular systems.

The unknown input problem (UIO) involves the estimation of the state of a dynamic system subject to unknown input excitations [5]. Many published works focus on the issue of simultaneously estimating the system state and the unknown input vectors in the presence of unknown disturbances. Specifically, UIOs are involved in practical engineering systems, such as diagnosis and fault detection [6,7], secure communication [8,9], or systems where the measurement of the inputs is either practically impossible or too expensive [10,11], while an equivalent input observer is often required to estimate the exerting force or torque of a robotic system or the cutting force of a machine tool [11]. Other engineering systems, such as industrial biological processes, wastewater treatment processes, and fuel cell stack systems, can be found in [11] and references therein.

For a strictly proper system, different approaches are given in [5,12,13] to simultaneously estimate the state and unknown input disturbances. However, they are restricted to particular assumptions. For example, it is assumed in [5] that the number of independent signals $l$ in the unknown input disturbance $d(t)$ must be no greater than the output dimension and that the matrix of unknown inputs $G_{n \times l}$ (given as $d(t)_{n \times 1} = G_{n \times l} \bar{d}(t)_{l \times 1}$) and some pre-specified rank conditions must be known. Notice that it is more suitable to estimate the net equivalent input disturbance (EID) of the unknown mismatched input disturbance than to estimate the disturbance itself, because the control input is required to suppress the negative effects of the mismatched input disturbance. Regardless, whenever the output number $p < l$ or the input number $m < l$, it is impossible to separately estimate the unknown mismatched input disturbance. As an alternative to separately estimating input or output disturbances, as in most of the existing UIO design approaches presented recently, in this paper it is proposed to estimate the net EID of the system.

Based on the proof of the existence of EID for strictly proper systems [14], an EID is developed in Section 3 of this paper as an extension to the case of proper continuous-time systems. Given a controllable and observable proper system with all stable zeros and unknown mismatched independent input disturbances $d_i(t) = G_i \bar{d}_i(t)$ and output disturbances $d_o(t) = G_o \bar{d}_o(t)$, an EID $d_e(t)$ exists that enters the plant through the control input channel $B$ and the input–output direct feedthrough term $D$, such that the two systems—one with $\{d_i(t) \text{ and } d_o(t)\}$ and the other with the matched disturbances $\{Bd_e(t) \text{ and } Dd_e(t)\}$—share the same outputs but differ in their states [15,16].

The simultaneous estimation of the system state and the EID of a strictly proper regular system subject to mismatch-dependent input and output disturbances was discussed in [11]. Based on the advanced functional observer originally presented in [10] for strictly proper regular systems, a modified functional observer for regular systems with an input–output direct feedthrough term is developed in Section 4 to estimate the net EID under the effects of disturbances. The EID method in conjunction with the modified functional observer can deal with proper systems with mismatch and independent input and output disturbances, i.e., $d_i(t) = G_i\bar{d}_i(t)$ and $d_o(t) = G_o\bar{d}_o(t)$, where $G_i \in \mathfrak{R}^{n \times l_i}$ and $G_o \in \mathfrak{R}^{p \times l_o}$ are unknown and there is no additional constraint for $p \geq l = \max(l_i, l_o)$.

The main contributions of this paper can be briefly discussed as follows. Consider a practical singular analog system in which the input and output are measurable. From a theoretical point of view, no existing MATLAB toolbox can be used to simulate the states of the singular analog system to construct the outputs, even if a mathematical model of the singular analog system $(E_r, A_r, B_r, C_r)$ is available, due to the singular matrix $E_r$ in the term $E_r\dot{x}(t)$. In addition, an approach for directly solving the tracker-based generalized algebraic Riccati equation for the singular analog system $(E_r, A_r, B_r, C_r)$ has not yet been developed in the literature. Using the proposed equivalent reduced-order proper regular model, these issues can be indirectly overcome, and also the controller can be designed. Now, since no mathematical models exist for unknown singular analog systems, the off-line observer–Kalman-filter identification (OKID) method [17] can be applied to construct the aforementioned equivalent reduced-order proper regular model for the singular system. As a result, in this paper, such a method is utilized to construct the newly developed modified functional observer (using the system output to estimate the integrated control input and estimated EID) and the EID estimator (to estimate the EID), such that the desired control input can consequently be determined. The developed equivalent system can also be utilized to directly realize the observer, and the control objective is then to estimate and feed the EID back through the control input channels to counteract the negative effects induced by the input and output disturbances, without any of the aforementioned constraints on the disturbances. This paper also provides more details on improving the disturbance rejection performance based on a net EID estimation approach for a system with $l_i$ unknown input disturbances and $l_o$ output disturbances imposed simultaneously, where $l_i, l_o > p$, with $p$ representing the number of outputs.

The rest of this paper is organized as follows. Preliminary findings of the optimal linear quadratic tracker design for singular systems and the motivation of this paper are given in Section 2. The properties of the EID associated with the proof for a proper continuous-time system with unknown input and output disturbances are given in Section 3. Section 4 presents the modified linear functional observer and the EID estimator, while Section 5 explains the design procedure for an OKID-based linear quadratic analog tracker (LQAT) associated with the modified functional observer for unknown non-square continuous-time singular systems. Illustrative examples are given in Section 6 to show the superiority of the proposed method. Lastly, conclusions are presented in Section 7.

## 2. Preliminaries and Motivation

*Model Conversion for Singular Systems*

Consider a known linear continuous-time singular system described by

$$E_r\dot{x}_c(t) = A_rx_c(t) + B_ru_c(t), \tag{1a}$$

$$y_c(t) = C_rx_c(t), \tag{1b}$$

where $x_c(t) \in \mathfrak{R}^n$, $u(t) \in \mathfrak{R}^m$, and $y_c(t) \in \mathfrak{R}^p$ are the state, control input, and measured output vectors of the system, respectively; $A_r \in \mathfrak{R}^{n \times n}$ and $B_r \in \mathfrak{R}^{n \times m}$ are real constant matrices; and $E_r \in \mathfrak{R}^{n \times n}$ is a singular matrix. For simplicity, it is assumed here that the system has no impulsive modes in the fast state. For a singular system with impulsive modes in the fast state, the methodology for

eliminating the impulsive modes can be found in [2]. Through a series of coordinate transforms, the impulsive mode-free singular system can be decomposed into a reduced-order regular subsystem and a non-dynamic subsystem [2], as

$$
\begin{bmatrix} I_\xi & 0 \\ 0 & 0 \end{bmatrix} \begin{bmatrix} \dot{\hat{x}}_{c,s}(t) \\ \dot{\hat{x}}_{c,f}(t) \end{bmatrix} = \begin{bmatrix} \hat{A}_s & 0 \\ 0 & I_{n-\xi} \end{bmatrix} \begin{bmatrix} \hat{x}_{c,s}(t) \\ \hat{x}_{c,f}(t) \end{bmatrix} + \begin{bmatrix} \hat{B}_s \\ \hat{B}_f \end{bmatrix} u_c(t), \tag{2a}
$$

$$
y_c(t) = \begin{bmatrix} \hat{C}_s & \hat{C}_f \end{bmatrix} \begin{bmatrix} \hat{x}_{c,s}(t) \\ \hat{x}_{c,f}(t) \end{bmatrix}, \tag{2b}
$$

where $\hat{x}_{c,f}(t) = -\hat{B}_f u_c(t)$. Then, the corresponding reduced-order regular system with an input–output feedthrough term can be described by

$$
\dot{\hat{x}}_{c,s}(t) = \hat{A}_s \hat{x}_{c,s}(t) + \hat{B}_s u_c(t), \tag{3a}
$$

$$
y_c(t) = \hat{C}_s \hat{x}_{c,s}(t) + (-\hat{C}_f \hat{B}_f) u_c(t) = \hat{C}_s \hat{x}_{c,s}(t) + \hat{D}_s u_c(t), \tag{3b}
$$

where $\hat{D}_s = -\hat{C}_f \hat{B}_f$, $\hat{A}_s \in \mathcal{R}^{\xi \times \xi}$, $\hat{B}_s \in \mathcal{R}^{\xi \times m}$, $\hat{C}_s \in \mathcal{R}^{p \times \xi}$, and $\hat{D}_s \in \mathcal{R}^{p \times m}$ are the state, input, output, and direct feedthrough matrices, respectively; $\hat{x}_{c,s}(t) \in \mathcal{R}^\xi$ is the state vector, $u_c(t) \in \mathcal{R}^m$ is the control input, and $y_c(t) \in \mathcal{R}^p$ is the measurable output of the system at time $t$.

For this equivalent model, let the associated cost function to be minimized be

$$
J(y_c(t), u_c(t)) = \frac{1}{2} \int_0^{t_f} \left\{ [y_c(t) - r(t)]^T Q_c [y_c(t) - r(t)] + u_c(t)^T R_c u_c(t) \right\} dt, \tag{4}
$$

where $Q_c$ is a $p \times p$ positive definite or positive semi-definite real symmetric matrix, $R_c$ is an $m \times m$ positive definite real symmetric matrix, the reference input $r(t)$ denotes the pre-specified output trajectory, and the final index is finite, i.e., $t_f < \infty$. Solving Equation (4) yields the continuous-time state feedback control law [18]

$$
\begin{aligned}
u_c(t) &= -K_c \hat{x}_{c,s}(t) - \overline{R}_c^{-1} \left[ \left( \hat{C}_s - \hat{D}_s K_c \right) \left( \hat{A}_s - \hat{B}_s K_c \right)^{-1} \hat{B}_s + \hat{D}_s \right]^T Q_c r(t) \\
&= -K_c \hat{x}_{c,s}(t) + E_c r(t),
\end{aligned} \tag{5}
$$

where

$$
K_c = \overline{R}_c^{-1} \left( \hat{B}_s^T P_s + N_s^T \right), \; E_c = -\overline{R}_c^{-1} \left[ \left( \hat{C}_s - \hat{D}_s K_c \right) \left( \hat{A}_s - \hat{B}_s K_c \right)^{-1} \hat{B}_s + \hat{D}_s \right]^T Q_c,
$$
$$
\overline{R}_c = R_c + \hat{D}_s^T Q_c \hat{D}_s, \; N_s = \hat{C}_s^T Q_c \hat{D}_s,
$$

and $P_s > 0$ satisfies the algebraic Riccati equation

$$
\hat{A}_s^T P_s + P_s \hat{A}_s - \left( P_s \hat{B}_s + N_s \right) \overline{R}_c^{-1} \left( \hat{B}_s^T P_s + N_s^T \right) + \hat{C}_s^T Q_c \hat{C}_s = 0. \tag{6}
$$

Our previous work [4] shows that theoretically, the tracker-design-oriented algebraic Riccati equation (ARE) for a regular system can be directly generalized for (i) a singular system in terms of $(E_r, A_r, B_r, C_r)$ in Equations (1a) and (1b); (ii) its equivalent model in terms of $(\hat{E}, \hat{A}, \hat{B}, \hat{C})$, with a non-symmetric singular matrix $\hat{E}$ for the singular system with impulsive modes, where $\hat{E} = \begin{bmatrix} I_\xi & 0 \\ 0 & \hat{E}_f \end{bmatrix}_{n \times n}$, $\hat{A} = \begin{bmatrix} \hat{A}_s & 0 \\ 0 & I_{n-\xi} \end{bmatrix}_{n \times n}$, $\hat{B} = \begin{bmatrix} \hat{B}_s \\ \hat{B}_f \end{bmatrix}_{n \times m}$, $\hat{C} = \begin{bmatrix} \hat{C}_s & \hat{C}_f \end{bmatrix}_{p \times n}$; or (iii) its equivalent model in terms of $(\hat{E}, \hat{A}, \hat{B}, \hat{C})$, with a symmetric singular matrix $\hat{E}$ for the impulsive-mode-free singular system, where $\hat{E} = \begin{bmatrix} I_\xi & 0 \\ 0 & 0_{n-\xi} \end{bmatrix}_{n \times n}$. It is worthwhile noting that if $\hat{C}_f \neq 0$, then the generalized Riccati equation for a singular system might have no solution $P \in \mathcal{R}^{n \times n}$, even if $Q_c \in \mathcal{R}^{p \times p}$, and $R_c \in \mathcal{R}^{m \times m}$ are chosen as positive definite matrices.

For the known linear continuous-time singular system described in Equations (1a) and (1b), through a series of coordinate transforms, the impulsive-mode-free singular system can be decomposed into a reduced-order regular subsystem with a direct input–output feedthrough term $\hat{D}_s$, described in Equations (3a) and (3b). As a result, the corresponding Riccati equation in Equation (6) is solvable; consequently, an optimal tracker for the singular system can be designed. As for the unknown linear singular system, the aforementioned series of transforms are not available. To overcome this issue, we are motivated to take the merits of the OKID method to directly identify the equivalent discrete-time reduced-order regular subsystem with a direct input–output feedthrough term in the general coordinate form for the impulsive mode-free singular system. Then, we obtain the corresponding continuous-time model through the discrete-to-continuous model conversion. With this approach, the aforementioned series of transforms can be avoided, even for the given linear continuous-time singular system.

For the unknown square–non-square singular sampled data systems subject to unknown input and output disturbances, a new functional observer-based discrete equivalent input disturbance (EID) estimator was presented in our previous work [4]. The main objective of this paper is to propose a new functional observer-based EID estimator for the unknown square–non-square singular continuous-time systems subject to unknown input and output disturbances.

## 3. Property of the EID Estimator

Consider a continuous-time controllable and observable system with stable zeros and unknown disturbances, described by

$$\dot{x}_c(t) = Ax_c(t) + Bu_c(t) + G_i\bar{d}_i(t), \tag{7a}$$

$$y_c(t) = Cx_c(t) + Du_c(t) + G_o\bar{d}_o(t), \tag{7b}$$

where $x_c(t) \in \mathcal{R}^n$, $u_c(t) \in \mathcal{R}^m$, and $y_c(t) \in \mathcal{R}^p$, respectively, represent the state, input, and output vectors; $A \in \mathcal{R}^{n\times n}$, $B \in \mathcal{R}^{n\times m}$, $C \in \mathcal{R}^{p\times n}$, and $D \in \mathcal{R}^{p\times m}$ are known matrices; $\bar{d}_i(t) \in \mathcal{R}^{l_i}$ and $\bar{d}_o(t) \in \mathcal{R}^{l_o}$ are the mismatched input and output disturbance vectors, respectively; $G_i \in \mathcal{R}^{n\times l_i}$ and $G_o \in \mathcal{R}^{p\times l_o}$ are matrices. Here, $G_i$, $G_o$, $\bar{d}_i(t)$ and $\bar{d}_o(t)$ are assumed unknown. Let matrices $B$ and $C$ be full column and full row ranks, respectively, and $\text{rank}(CB) = p \leq m \leq n$. There is no constraint on the number of unknown inputs in $\bar{d}_i(t)$, i.e., there may be more unknown inputs than control inputs and measured outputs, and also the disturbances may be state-dependent. Section 9.5 in [19] presents a discussion on the effects of state-dependent uncertainties appearing in the process model as either additive or multiplicative disturbances (which can be included in a sensitivity analysis of combined estimation and control) and the associated algorithms. Additionally, other robust control approaches concerning state-dependent disturbances can be found in [11,20].

It is well-known that it is difficult or even impossible to estimate the components of $\bar{d}_i(t)$ and $\bar{d}_o(t)$ in a precise manner. It is also known that in most disturbed servo systems, the unknown inputs are estimated for disturbance rejection purposes to maintain the desired performance. An alternative is to assume the existence of a net EID, $d_e(t) \in R^m$, of the unknown disturbances $\bar{d}_i(t)$ and $\bar{d}_o(t)$ entering the plant through the control input channel $B$ and the direct feedthrough term $D$, meaning Equations (7a) and (7b) can be expressed as

$$\dot{x}_e(t) = Ax_e(t) + B[u_c(t) + d_e(t)], \tag{8a}$$

$$y_e(t) = Cx_e(t) + D[u_c(t) + d_e(t)], \tag{8b}$$

where $x_e(t) \in \mathcal{R}^n$ and $y_e(t) \in \mathcal{R}^p$ are the equivalent state and output vectors, respectively, with $x_e(0) = x_c(0)$. Notice that $x_e(t) \neq x_c(t)$ for $t > 0$, because in general $Bd_e(t) \neq G_i\bar{d}_i(t)$, even for $d_o(t) = 0$.

Similar to the case of a strictly proper system (i.e., no $Du_c(t)$ term) [14], the definition of net EID of an input–output feedthrough system with both mismatched input and output disturbances is analogous to the definition given in [4].

**Definition 1.** *Let the control inputs of Equations (7a) and (7b), (8a) and (8b) be $u_c(t) = 0$; let the output of Equations (7a) and (7b) with mismatched disturbances $G_i \overline{d}_i(t)$ and $G_o \overline{d}_o(t)$ be $y_c(t)$; and let the output of Equations (8a) and (8b) with matched disturbance $d_e(t)$ be $y_e(t)$. The disturbance $d_e(t)$ is known as the net equivalent input disturbance (EID) of the mismatched disturbances $G_i \overline{d}_i(t)$ and $G_o \overline{d}_o(t)$ if $y_e(t) = y_c(t)$, with $x_e(0) = x_c(0)$ for $t \geq 0$.*

It is important to remark that $y_e(t) = y_c(t)$ does not imply that $x_e(t) = x_c(t)$, because given $y_c(t)$ and the output matrix $C \in \mathcal{R}^{p \times n}$ with $p < n$, $y_c(t)$ will be equal to $C_c x_c(t)$ for an infinite number of $x_c(t)$. Now, since Equations (8a) and (8b) can be used as equivalents of Equations (7a) and (7b), the disturbance rejection of the servo problem can be formulated as an estimate and can feed $d_e(t)$ back to the input terminal to cancel the negative effects of $G_i \overline{d}_i(t)$ and $G_o \overline{d}_o(t)$.

On the other hand, the proof of the theoretical guarantee of the existence of the meaningful net EID is analogous to the proof presented in [4], and hence it is omitted here.

## 4. Modified Functional Observer with Unknown Input

Considering an unknown non-square continuous-time impulsive-mode-free singular system with more control input channels than output channels and all stable control zeros, it is desirable to propose a high-performance optimal analog state estimate tracker with an EID estimator for the system, with unknown matched and mismatched input and output disturbances, as shown in Figure 1. The newly developed modified functional observer uses the system output to estimate the integrated control input and estimated EID, while the EID estimator estimates the EID such that the desired control input can be determined consequently. It is worthwhile noting that for effective determination of the EID estimator in the conventional method, it is required to have the known control input vector with unknown disturbances. To overcome this constraint problem, in this paper the conventional EID estimator is extended for a system with both an unknown control input vector and unknown disturbances. Then, the aforementioned mechanism can be implemented as depicted in Figure 1.

In Chapter 7 of [11], the system of interest has dependent input and output disturbances, where the same disturbance $d(t)$ appears in both input and output terminals, and has no direct feedthrough term, as below

$$\dot{x}_c(t) = A x_c(t) + B u(t) + G_i d(t), \tag{9a}$$

$$y_c(t) = C x_c(t) + G_o d(t), \tag{9b}$$

where $x_c(t) \in \mathcal{R}^n$, $y_c(t) \in \mathcal{R}^p$, and $u(t) \in \mathcal{R}^m$ are the state, measured output, and input vectors, respectively; and $A \in \mathcal{R}^{n \times n}$, $B \in \mathcal{R}^{n \times m}$, $C \in \mathcal{R}^{p \times n}$, $G_i \in \mathcal{R}^{n \times l}$, $G_o \in \mathcal{R}^{p \times l}$ are known real constant matrices. The unknown disturbance $d(t) \in \mathcal{R}^l$ directly affects both the state and output of the system, and it is also assumed that rank$(C) = p$ and $p \geq l$.

Based on the advanced functional observer originally presented for strictly proper systems, in this section, a modified functional observer for a system with an input–output direct feedthrough term is derived to estimate the EID, which can be used to deal with proper systems with mismatch and independent input and output disturbances, i.e., $d_i(t) = G_i \overline{d}_i(t)$ and $d_o(t) = G_o \overline{d}_o(t)$, where $G_i \in \mathcal{R}^{n \times l_i}$ and $G_o \in \mathcal{R}^{p \times l_o}$ are assumed to be unknown and there is no more constraint for $p \geq l = \max(l_i, l_o)$.

### 4.1. Problem Statement

Consider a linear proper system given as

$$\dot{x}_c(t) = A x_c(t) + B u_f(t), \tag{10a}$$

$$y_c(t) = C x_c(t) + D u_f(t), \tag{10b}$$

where $A \in \mathcal{R}^{n \times n}$, $B \in \mathcal{R}^{n \times m}$, $C \in \mathcal{R}^{p \times n}$, and $D \in \mathcal{R}^{p \times m}$ are known matrices, and the vector $u_f(t) = u_c(t) + d_e(t) \in \mathcal{R}^m$ is the unknown input vector, which affects both the state and the output of the system. It is assumed that $\text{rank}(C) = p$ and $p \geq m$. The objective of the reduced-order linear functional observer is to estimate a linear combination of the state or the unknown input vector.

We define the augmented state vector $\omega(t) = \begin{bmatrix} x_e^T(t) & u_f^T(t) \end{bmatrix}^T \in \mathcal{R}^{(n+m)}$ such that Equations (10a) and (10b) can be expressed as

$$\overline{E}_\omega \dot{\omega}(t) = \overline{A}_\omega \omega(t), \tag{11a}$$

$$y_c(t) = \overline{C}_\omega \omega(t), \tag{11b}$$

where $\overline{E}_\omega = \begin{bmatrix} I_{n \times n} & 0_{n \times m} \\ 0_{m \times n} & 0_{m \times m} \end{bmatrix} \in \mathcal{R}^{(n+m) \times (n+m)}$, $\overline{A}_\omega = \begin{bmatrix} A & B \\ 0_{m \times n} & 0_{m \times m} \end{bmatrix} \in \mathcal{R}^{(n+m) \times (n+m)}$, and $\overline{C}_\omega = \begin{bmatrix} C & D \end{bmatrix} \in \mathcal{R}^{p \times (n+m)}$. We define the functional state vector $z(t) \in \mathcal{R}^\kappa$ that must be reconstructed (or estimated) as

$$z(t) = F\omega(t) = \begin{bmatrix} F_1 & 0 \\ 0 & F_2 \end{bmatrix} \begin{bmatrix} x_e(t) \\ u_f(t) \end{bmatrix}, \tag{12}$$

where $F_1 \in \mathcal{R}^{(\kappa-m) \times n}$, $F_2 \in \mathcal{R}^{m \times m}$, and $F \in \mathcal{R}^{\kappa \times (n+m)}$ are given constant matrices that satisfy $\text{rank}(F) = \kappa$ and $\text{rank} \begin{bmatrix} \overline{C}_\omega \\ F \end{bmatrix} = (p + \kappa) \leq (n + m)$.

A reduced-order observer will be now proposed to estimate $z(t)$. Consider the observer structure of order $\kappa$

$$\dot{w}(t) = Nw(t) + Jy_c(t), \tag{13a}$$

$$\hat{z}(t) = w(t) + Qy_c(t), \tag{13b}$$

for Equations (10a) and (10b), where $w(t) \in \mathcal{R}^\kappa$, $\hat{z}(t)$ is the estimate of $z(t)$, and matrices $N$, $J$ and $Q$ should be determined, such that $\hat{z}(t)$ converges asymptotically to $z(t)$, i.e., $\hat{z}(t) \to z(t)$, as $t \to \infty$.

With the linear functional state vector defined in Equation (12), the proposed observer in Equations (14a) and (14b) offers great flexibility in estimating any linear combination of the state and the unknown input of the system in Equations (10a) and (10b). For instance, a linear combination of only the unknown input $u_f(t)$ or only the states to be estimated can be used, by setting $F_1 = 0_{(\kappa-m) \times n}$ or $F_2 = 0_m$, respectively. In other words, it can be assumed that $F = \begin{bmatrix} F_1 & 0 \\ 0 & I_m \end{bmatrix}$, where $rank(F_1) = \kappa - m$ is used to fulfill the rank restriction of $F$. Thus, a linear combination of the estimated state $\hat{x}_e(t)$ and the estimated unknown input $\hat{u}_f(t)$ can be obtained simultaneously. Consequently

$$\hat{z}(t) = F\hat{\omega}(t) = \begin{bmatrix} F_1\hat{x}_e(t) \\ \hat{u}_f(t) \end{bmatrix} \tag{14}$$

and

$$\hat{u}_f(t) = \hat{u}_c(t) + \hat{d}_e(t) = \begin{bmatrix} 0_{m \times (\kappa-m)} & I_m \end{bmatrix} \hat{z}(t), \tag{15}$$

which implies $\hat{d}_e(t) = \begin{bmatrix} 0_{m \times (\kappa-m)} & I_m \end{bmatrix} \hat{z}(t) - \hat{u}_c(t) = \begin{bmatrix} 0_{m \times (\kappa-m)} & I_m \end{bmatrix} \hat{z}(t) - u_c(t)$.

It is worthwhile to notice that $u_f(t) = u_c(t) + d_e(t) = -K_c\hat{x}_e(t) + E_c r(t)$, where $K_c$ and $E_c$ are determined based on [18] and the system is fictitiously considered to be disturbance-free, since the net EID $d_e(t)$ of the unknown input and output disturbances as well as the tracking errors have been theoretically merged to the control input terminal. To determine the control input $u_c(t)$, it is required to first estimate the EID $\hat{d}_e(t)$. However, to obtain the estimated EID $\hat{d}_e(t)$, it is required to first obtain the control input $u_c(t)$. To overcome this causality problem, we approximate the control input as $u_c(t) \cong \hat{u}_c(t) = \hat{u}_f(t) - \hat{d}_e(t)$, which implies $\hat{d}_e(t) = \hat{u}_f(t) - u_c(t) = \begin{bmatrix} 0_{m \times (\kappa-m)} & I_m \end{bmatrix} \hat{z}(t) - u_c(t)$, where $\hat{u}_f(t) = \begin{bmatrix} 0_{m \times (\kappa-m)} & I_m \end{bmatrix} \hat{z}(t)$. The aforementioned causality mechanism can then be implemented as depicted in Figure 1.

Notice that the augmented state vector $\omega(t) = [\; x_e^T(t) \quad d^T(t) \;]^T \in \mathcal{R}^{(n+l)}$ in the original functional observer [11] is now modified to $\omega(t) = [\; x_e^T(t) \quad u_f^T(t) \;]^T \in \mathcal{R}^{(n+m)}$ in this paper. This proposed modified functional observer compensates for the system disturbance, even if $G_i$ and $G_o$ are unknown. Since in models of the real physical system $G_i$ and $G_o$ are usually unknown, it is more realistic to estimate the equivalent unknown input instead of the disturbances directly.

### 4.2. Existence Conditions

As the functions to be utilized have been defined, this section presents the reconstruction of a linear function problem. Assuming that any unobservable state can be eliminated, by defining an observer state vector of a lower dimension, the order $\kappa$ of the observer defined in Equations (13a) and (13b) should be less than or equal to the reduced-order state observer, i.e., $\kappa \leq n + m - p$.

In Equation (13b), the output $\hat{z}(t)$ provides an asymptotic estimate of $F\omega(t)$ if

$$\lim_{t \to \infty} [\hat{z}(t) - F\omega(t)] = 0. \tag{16}$$

Given a full-row rank matrix $\mathcal{L} \in \mathcal{R}^{\kappa \times (n+m)}$, the two error vectors $\varepsilon(t) \in \mathcal{R}^{\kappa}$ and $e(t) \in \mathcal{R}^{\kappa}$ can be described by

$$\varepsilon(t) = w(t) - \mathcal{L}\overline{E}_\omega \omega(t), \tag{17}$$

$$e(t) = \hat{z}(t) - z(t), \tag{18}$$

where $w(t)$ in Equation (13a) estimates a different linear combination of $\omega(t)$. Then,

$$\begin{aligned}
\dot{\varepsilon}(t) &= \dot{w}(t) - \mathcal{L}\overline{E}_\omega \dot{\omega}(t) = (Nw(t) + Jy_c(t)) - \mathcal{L}(\overline{A}_\omega \omega(t)) \\
&= Nw(t) - N\mathcal{L}\overline{E}_\omega \omega(t) + N\mathcal{L}\overline{E}_\omega \omega(t) + J\overline{C}_\omega \omega(t) - \mathcal{L}\overline{A}_\omega \omega(t) \\
&= N\varepsilon(t) + (N\mathcal{L}\overline{E}_\omega + J\overline{C}_\omega - \mathcal{L}\overline{A}_\omega)\omega(t).
\end{aligned} \tag{19}$$

$$\begin{aligned}
e(t) &= \hat{z}(t) - z(t) = \hat{z}(t) - F\omega(t) = (w(t) + Qy_c(t)) - F\omega(t) \\
&= w(t) - \mathcal{L}\overline{E}_\omega \omega(t) + \mathcal{L}\overline{E}_\omega \omega(t) + Q\overline{C}_\omega \omega(t) - F\omega(t) \\
&= \varepsilon(t) + (\mathcal{L}\overline{E}_\omega + Q\overline{C}_\omega - F)\omega(t).
\end{aligned} \tag{20}$$

This introduces the following theorem.

**Theorem 1.** *From Equations (19) and (20), $\hat{z}(t)$ in (13b) is an asymptotic estimate of $z(t)$ if matrix $\mathcal{L} \in \mathcal{R}^{\kappa \times (n+m)}$ exists, such that the following equations*

$$N\mathcal{L}\overline{E}_\omega + J\overline{C}_\omega - \mathcal{L}\overline{A}_\omega = 0, \; N \text{ is Hurwitz} \tag{21}$$

*and*

$$\mathcal{L}\overline{E}_\omega + Q\overline{C}_\omega - F = 0 \tag{22}$$

*are satisfied [11]. Matrices $N$, $J$, and $Q$ are shown in Equations (13a) and (13b); and $\overline{E}_\omega$ is given in Equation (11a). Upon satisfaction of the conditions in Equations (21) and (22), the dynamic equation in Equation (19) can be reduced to $\dot{\varepsilon}(t) = N\varepsilon(t)$, which implies that $\varepsilon(t) \to 0$ as $t \to \infty$, and consequently $e(t) \to 0$.*

To derive the parameters of the linear functional observer (Equations (13a) and (13b)), it is necessary to solve matrix Equations (21) and (22) for the unknown matrices $N$(Hurwitz), $\mathcal{L}$, $Q$, and $J$. The following theorem provides necessary and sufficient conditions for the solvability of matrix Equations (21) and (22), while ensuring that matrix $N$ is a Hurwitz matrix

**Theorem 2.** *Matrix Equations (21) and (22) are completely solvable with, $N$ being a Hurwitz matrix, only if the following two conditions hold:*

***Condition 1.***

$$\text{rank}\begin{bmatrix} F\overline{A}_\omega & F \\ \overline{C}_\omega\overline{A}_\omega & \overline{C}_\omega \\ \overline{C}_\omega & 0 \\ 0 & \overline{E}_\omega \\ F & 0 \end{bmatrix} = \text{rank}\begin{bmatrix} \overline{C}_\omega\overline{A}_\omega & \overline{C}_\omega \\ \overline{C}_\omega & 0 \\ 0 & \overline{E}_\omega \\ F & 0 \end{bmatrix}. \tag{23}$$

***Condition 2.***

$$\text{rank}\begin{bmatrix} sF - F\overline{A}_\omega & -F \\ \overline{C}_\omega\overline{A}_\omega & \overline{C}_\omega \\ \overline{C}_\omega & 0 \\ 0 & \overline{E}_\omega \end{bmatrix} = \text{rank}\begin{bmatrix} \overline{C}_\omega\overline{A}_\omega & \overline{C}_\omega \\ \overline{C}_\omega & 0 \\ 0 & \overline{E}_\omega \\ F & 0 \end{bmatrix} \quad \forall s \in \mathbb{C}, \ \Re(s) \geq 0. \tag{24}$$

**Proof.** The part of the proof related to the design procedure is now briefly summarized. The complete proof is omitted here and can be obtained by appropriately modifying the ones in [11]. Through some mathematical manipulations, Equations (21) and (22) induce

$$NF = F\overline{A}_\omega - \begin{bmatrix} Q & T & \mathcal{L} \end{bmatrix}\begin{bmatrix} \overline{C}_\omega\overline{A}_\omega \\ \overline{C}_\omega \\ \widetilde{E}\overline{A}_\omega \end{bmatrix}. \tag{25}$$

More precisely, the proof can be obtained from Equation (21) by

(i) substituting $\widetilde{E} = \overline{E}_\omega - I_{(n+m)}$ into Equation (22) to get $\mathcal{L} = F - Q\overline{C}_\omega - \mathcal{L}\widetilde{E}$;

(ii) substituting Equation (22) into Equation (21) to get $N\mathcal{L}\overline{E}_\omega + J\overline{C}_\omega = N(F - Q\overline{C}_\omega) + J\overline{C}_\omega = \mathcal{L}\overline{A}_\omega$ and $NF = NQ\overline{C}_\omega - J\overline{C}_\omega + \mathcal{L}\overline{A}_\omega = (-T)\overline{C}_\omega + \mathcal{L}\overline{A}_\omega$, where $T = (J - NQ)$;

(iii) using (i) and (ii) yields $NF = -T\overline{C}_\omega + (F - Q\overline{C}_\omega - \mathcal{L}\widetilde{E})\overline{A}_\omega = F\overline{A}_\omega - \begin{bmatrix} Q & T & \mathcal{L} \end{bmatrix}\begin{bmatrix} \overline{C}_\omega\overline{A}_\omega \\ \overline{C}_\omega \\ \widetilde{E}\overline{A}_\omega \end{bmatrix}.$

□

Since $F$ is a known matrix, we define the full-row rank matrix

$$\begin{bmatrix} H_1 & E_1 \end{bmatrix} = \begin{bmatrix} F^+ & (I_{n+m} - F^+F) \end{bmatrix}, \tag{26}$$

where $F^+$ is the Moore–Penrose inverse of $F$, where $FH_1 = I_\kappa$ and $FE_1 = 0_{\kappa \times (n+m)}$. Post-multiplying both sides of Equation (25) by Equation (26) gives

$$N = F\overline{A}_\omega H_1 - \begin{bmatrix} Q & T & \mathcal{L} \end{bmatrix}\begin{bmatrix} \overline{C}_\omega\overline{A}_\omega H_1 \\ \overline{C}_\omega H_1 \\ \widetilde{E}\overline{A}_\omega H_1 \end{bmatrix}, \tag{27}$$

where $T = (J - NQ)$ and

$$F\overline{A}_\omega E_1 = \begin{bmatrix} Q & T & \mathcal{L} \end{bmatrix}\begin{bmatrix} \overline{C}_\omega\overline{A}_\omega E_1 \\ \overline{C}_\omega E_1 \\ \widetilde{E}\overline{A}_\omega E_1 \end{bmatrix}. \tag{28}$$

In Equation (28), $F$, $\overline{A}_\omega$, $E_1$, $\overline{C}_\omega$, and $\widetilde{E}$ are known matrices, while matrices $Q$, $T$, and $\mathcal{L}$ are unknown and need to be determined to find matrix $N$ in Equation (27).

Now, we augment Equation (28) with Equation (22) as

$$\begin{bmatrix} Q & T & \mathcal{L} \end{bmatrix}\Delta = \psi, \tag{29}$$

where $\Delta \in \mathfrak{R}^{(2p+n+m)\times(2n+2m)}$ and $\psi \in \mathfrak{R}^{\kappa\times(2n+2m)}$ are known matrices defined by

$$\Delta = \begin{bmatrix} \overline{C}_\omega \overline{A}_\omega E_1 & \overline{C}_\omega \\ \overline{C}_\omega E_1 & 0 \\ \widetilde{EA}_\omega E_1 & \overline{E}_\omega \end{bmatrix} \tag{30a}$$

and

$$\psi = \begin{bmatrix} F\overline{A}_\omega E_1 & F \end{bmatrix}. \tag{30b}$$

A necessary and sufficient condition for the existence of a solution $\{Q, T, \text{ and } \mathcal{L}\}$ can be derived from Equation (29). Then, a necessary and sufficient condition for ensuring that matrix $N$ is a Hurwitz function can be derived by substituting $Q$, $T$, and $\mathcal{L}$ into Equation (27). At last, the matrix $J$ is obtained as $J = T + NQ$. As a result, the unknown matrices $N, \mathcal{L}, J, Q$ and $F$ satisfy matrix Equations (21) and (22) of Theorem 1.

Based on the general solution of the linear matrix equations [21], a solution to Equation (29) exists only if

$$\text{rank}\begin{bmatrix} \psi \\ \Delta \end{bmatrix} = \text{rank}(\Delta),$$

meaning that

$$\text{rank}\begin{bmatrix} F\overline{A}_\omega E_1 & F \\ \overline{C}_\omega \overline{A}_\omega E_1 & \overline{C}_\omega \\ \overline{C}_\omega E_1 & 0 \\ \widetilde{EA}_\omega E_1 & \overline{E}_\omega \end{bmatrix} = \text{rank}\begin{bmatrix} \overline{C}_\omega \overline{A}_\omega E_1 & \overline{C}_\omega \\ \overline{C}_\omega E_1 & 0 \\ \widetilde{EA}_\omega E_1 & \overline{E}_\omega \end{bmatrix}. \tag{31}$$

It can be proved that Condition 1 of Theorem 2 and Equation (31) are equivalent (see Appendix A.1) by post-multiplying both sides of Equation (23) by the full-row rank matrix $\begin{bmatrix} H_1 & E_1 & 0 \\ 0 & 0 & I_{n+m} \end{bmatrix}$. Therefore, upon satisfaction of Condition 1 of Theorem 2 and using the generalized matrix inverse approach [21], a solution to Equation (29) always exists and is given by

$$\begin{bmatrix} Q & T & \mathcal{L} \end{bmatrix} = \psi\Delta^+ + Z(I_{2p+n+m} - \Delta\Delta^+), \tag{32}$$

where $\Delta^+$ is the generalized inverse of $\Delta$, and $Z \in \mathfrak{R}^{\kappa\times(2p+n+m)}$ is an arbitrary matrix, which will be further utilized to establish the stability of matrix $N$.

Then, substituting Equation (32) into Equation (27) results in

$$N = \Phi - Z\Omega, \tag{33}$$

where $\Phi \in \mathfrak{R}^{\kappa\times\kappa}$ and $\Omega \in \mathfrak{R}^{(2p+n+m)\times\kappa}$ are known matrices, which can be defined as

$$\Phi = (F\overline{A}_\omega H_1 - \psi\Delta^+\widetilde{\Theta}), \quad \Omega = (I_{2p+n+m} - \Delta\Delta^+)\widetilde{\Theta}, \tag{34}$$

in which $\widetilde{\Theta} = \begin{bmatrix} \overline{C}_\omega \overline{A}_\omega H_1 \\ \overline{C}_\omega H_1 \\ \widetilde{EA}_\omega H_1 \end{bmatrix}$. In Equation (33), the matrix $N$ is a Hurwitz function for matrix $Z$ only if the pair $(\Omega, \Phi)$ is detectable, i.e.,

$$rank\begin{bmatrix} sI_\kappa - \Phi \\ \Omega \end{bmatrix} = \kappa, \ \forall s \in \mathbb{C}, \ \mathfrak{R}(s) \geq 0. \tag{35}$$

Now, it can be shown that Condition 2 of Theorem 2 is equivalent to Equation (35) by post-multiplying both sides of Equation (24) by the full-row matrix $\begin{bmatrix} H_1 & E_1 & 0 \\ 0 & 0 & I_{n+m} \end{bmatrix}$ (see Appendix A.2), therefore ensuring that matrix $N$ is a Hurwitz function.

**Remark 1.** *As in [4], the modified functional observer proposed here is only applicable for proper systems and not for strictly proper systems, because if $D = 0_{p \times m}$, i.e., $\overline{C}_\omega = [\; C \quad 0 \;]$, the column rank of the existing condition (Equation (23)) drops under the assumption that the system is square and $F$ is of full-row rank $(rank(F) = \kappa)$. An alternative method for strictly proper systems can be found in [14], however it does not apply to proper systems. This remark can be proven, as shown in [4].*

### 4.3. State–Space Structure of the Observer and the EID Estimator

To cancel the negative effects of the unknown inputs, the estimated EID $\hat{d}_e(t)$ can be newly constructed as

$$\hat{d}_e(t) = \hat{u}_f(t) - u_c(t), \tag{36}$$

where $\hat{u}_f(t)$ is the estimate of the system input $u_f(t) = u_c(t) + d_e(t)$. To achieve this goal, $\hat{u}_f(t)$ and $\hat{d}_e(t)$ are first estimated using the aforementioned extended method. Then, the desired control input $u_c(t)$ is obtained by filtering the estimated EID through a well-designed low-pass filter.

The accessorial variable $\widetilde{d}_e(t)$ is updated in real-time by filtering $\hat{d}_e(t)$ with the low-pass filter of the order $n_f (\geq m)$, described by

$$\dot{x}_f(t) = A_f x_f(t) + B_f \hat{d}_e(t), \tag{37a}$$

$$\widetilde{d}_e(t) = C_f x_f(t), \tag{37b}$$

where $A_f \in \mathcal{R}^{n_f \times n_f}$, $B_f \in \mathcal{R}^{n_f \times m}$, and $C_f \in \mathcal{R}^{m \times n_f}$ are the system, input, and output matrices, respectively.

Notice that Equations (37a) and (37b) gives a general input–output form, in which $x_f(t)$ does not have any direct physical implication and $n_f$ depends on the particular structure of this subsystem, which depends on the application. Furthermore, it is necessary to take $d_e(t)$ into account to estimate the system state $\hat{x}_e(t)$, and the state observer can be constructed as

$$\dot{\hat{x}}_e(t) = A\hat{x}_e(t) + Bu_f(t) + L_c[y_c(t) - \hat{y}_c(t)], \tag{38a}$$

$$\hat{y}_c(t) = C\hat{x}_e(t) + Du_f(t), \tag{38b}$$

where $\hat{x}_e(t) \in \mathcal{R}^n$ and $\hat{y}_c(t) \in \mathcal{R}^p$ are the estimated state and output vectors, respectively, and $L_c \in \mathcal{R}^{n \times p}$ is the estimation gain. With the estimated state $\hat{x}_e(t)$, the control input in Equations (7a) and (7b) is derived as

$$u_c(t) = u_f(t) - \widetilde{d}_e(t), \tag{39}$$

with

$$u_f(t) = -K_c \hat{x}_e(t) + E_c r(t), \tag{40}$$

where $K_c$ and $E_c$ are determined using Equation (5) and by considering the system to be disturbance-free, since the EID $d_e(t)$ of the unknown input and output disturbances as well as tracking errors have theoretically been merged to the control input terminal.

**Lemma 1** ([22,23]). *Let $(A, B)$ be the pair for the given open-loop system, and let $h \geq 0$ represent a prescribed degree of relative stability. Then, the eigenvalues of the closed-loop system $A - B\overline{R}_o^{-1}(BP + N_o^T)$ will lie to the left of the vertical line $-h$, with the matrix $P > 0$ being the solution of the Riccati equation*

$$(A + hI_n)^T P + P(A + hI_n) - (PB + N_o)\overline{R}_o^{-1}(B^T P + N_o^T) + Q_o = 0, \tag{41}$$

*where the matrix $I_n$ is the identity matrix, while $N_o = C^T Q_o D$, $\overline{R}_o = R_o + D^T Q_o D$, $R_o > 0$, and $Q_o \geq 0$ are weighting matrices with appropriate dimensions for the corresponding optimal regulator design.*

### 4.4. Stability Analysis in the Frequency Domain

Here, it is assumed that the only input to the system is the net EID. Therefore, from Equations (13a), (13b), (14), (37a), (37b), (38a), (38b), and (40), one has

$$\widetilde{d}_e(s) = C_f (sI_{n_f} - A_f)^{-1} B_f\, \hat{d}_e(s) \equiv G_{dd}(s)\hat{d}_e(s), \tag{42}$$

$$\hat{x}_e(s) = (sI_n - A + L_c C)^{-1} L_c y_c(s) + (sI_n - A + L_c C)^{-1}(B - L_c D)u_f(s), \tag{43}$$

$$u_f(s) = -K_c \hat{x}_e(s), \tag{44}$$

$$\hat{u}_f(s) = \begin{bmatrix} 0_{m\times(\kappa-m)} & I_m \end{bmatrix}\left[(sI_\kappa - N)^{-1}J + Q\right]y_c(s) \equiv G_{ufo}(s)y_c(s), \tag{45}$$

in the frequency domain, where $\widetilde{d}_e(s)$, $\hat{d}_e(s)$, $\hat{y}_c(s)$, $y_c(s)$, $u_f(s)$, $\hat{u}_f(s)$, and $\hat{x}_e(s)$ are the Laplace transforms of $\widetilde{d}_e(t)$, $\hat{d}_e(t)$, $\hat{y}_c(t)$, $y_c(t)$, $u_f(t)$, $\hat{u}_f(t)$, and $\hat{x}_e(t)$, respectively. Substituting Equation (44) into Equation (43) yields

$$\hat{x}_e(s) = (sI_n - A + L_c C)^{-1} L_c y_c(s) - (sI_n - A + L_c C)^{-1}(B - L_c D)K_c \hat{x}_e(s),$$
$$\left[I_n + (sI_n - A + L_c C)^{-1}(B - L_c D)K_c\right]\hat{x}_e(s) = (sI_n - A + L_c C)^{-1} L_c y_c(s),$$
$$\hat{x}_e(s) = \left[I_n + (sI_n - A + L_c C)^{-1}(B - L_c D)K_c\right]^{-1}(sI_n - A + L_c C)^{-1} L_c y_c(s) \equiv G_{xy}(s)y_c(s),$$

$$u_f(s) = -K_c \hat{x}_e(s) = -K_c G_{xy}(s)y_c(s), \tag{46}$$

where $G_{xy}(s) = \left[I_n + (sI_n - A + L_c C)^{-1}(B - L_c D)K_c\right]^{-1}(sI_n - A + L_c C)^{-1} L_c$.

Thus, from Equation (36) and Equation (39), the filtered disturbance estimate $\widetilde{d}_e(t)$ can be chosen as

$$\widetilde{d}_e(t) = f_d(t) \otimes \hat{d}_e(t) = f_d(t) \otimes (\hat{u}_f(t) - u_c(t)) = f_d(t) \otimes (\hat{u}_f(t) - u_f(t) + \widetilde{d}_e(t)), \tag{47}$$

where $\otimes$ denotes "convolution" and the Laplace transform of $f_d(t)$ is $G_{dd}(s)$, such that

$$\widetilde{d}_e(s) = G_{dd}(s)\hat{u}_f(s) - G_{dd}(s)u_f(s) + G_{dd}(s)\widetilde{d}_e(s), \tag{48}$$

$$\widetilde{d}_e(s) = (I_m - G_{dd}(s))^{-1}G_{dd}(s)(\hat{u}_f(s) - u_f(s)). \tag{49}$$

Substituting Equations (45) and (46) into Equation (49) yields

$$\widetilde{d}_e(s) = (I_m - G_{dd}(s))^{-1}G_{dd}(s)\left[G_{ufo}(s)y_c(s) - \left(-K_c G_{xy}(s)y_c(s)\right)\right]$$

$$= (I_m - G_{dd}(s))^{-1}G_{dd}(s)\left(G_{ufo}(s) + K_c G_{xy}(s)\right)y_c(s), \tag{50}$$

and thus

$$y_c(s) = \left[C(sI_n - A)^{-1}B + D\right](u_c(s) + d_e(s)) \equiv G_p(s)(u_c(s) + d_e(s)),$$

$$y_c(s) = G_p(s)\left(u_f(s) - \widetilde{d}_e(s) + d_e(s)\right), \tag{51}$$

where $G_p(s) = C(sI_n - A)^{-1}B + D$. Substituting Equations (46) and (50) into Equation (51) yields

$$y_c(s) = G_p(s)\left[-K_c G_{xy}(s)y_c(s)\right]$$
$$\qquad -G_p(s)\left[(I_m - G_{dd}(s))^{-1}G_{dd}(s)\left(G_{ufo}(s) + K_c G_{xy}(s)\right)y_c(s)\right] + G_p(s)d_e(s),$$

$$\left\{I_p + G_p(s)\left[K_c G_{yx}(s) + (I_m - G_{dd}(s))^{-1}G_{dd}(s)\left(G_{ufo}(s) + K_c G_{xy}(s)\right)\right]\right\}y_c(s) = G_p(s)d_e(s), \tag{52}$$

which, from Equation (52), implies that

$$
\begin{aligned}
y_c(s) &= \left\{ I_p + G_p(s)\left[ K_c G_{xy}(s) + (I_m - G_{dd}(s))^{-1} G_{dd}(s)\left( G_{ufo}(s) + K_c G_{xy}(s) \right) \right] \right\}^{-1} G_p(s) d_e(s) \\
&= \left[ I_p + G_p(s) K(s) \right]^{-1} G_p(s) d_e(s) \equiv G_{yd}(s) d_e(s),
\end{aligned}
\tag{53}
$$

where $G_p(s) = C(sI_n - A)^{-1}B + D$, $K(s) = K_c G_{xy}(s) + (I_m - G_{dd}(s))^{-1} G_{dd}(s)\left( G_{ufo}(s) + K_c G_{xy}(s) \right)$, $G_{yd}(s) = \left[ I_p + G_p(s) K(s) \right]^{-1} G_p(s)$ is the transfer function from the EID to the system output, and $I_m$, $I_n$, $I_p$, and $I_{n_f}$ are identity matrices of dimensions $m \times m$, $n \times n$, $p \times p$, and $n_f \times n_f$, respectively.

Note that Equation (53) is a multi-input-multi-output (MIMO) expression. To simplify the discussion, we denote $G_{yd}^{ij}(s)$ and $G_{dd}^{ij}(s)$ as the elements in the $i^{th}$ rows and $j^{th}$ columns of $G_{yd}(s)$ and $G_{dd}(s)$, respectively. From Equation (53), the effects of the unknown inputs can be effectively suppressed by minimizing $\|G_{yd}^{ij}(s)\|_\infty$ ($i = 1, \cdots, p$; $j = 1, \cdots, m$), and $\|G_{yd}^{ij}(s)\|_\infty$ is sufficiently small with appropriate design of $G_{dd}(s)$. For this case, the effect of the disturbances can be alleviated, since feedback into the system of the equivalent unknown inputs accurately estimated by the functional observer can effectively counteract actual unknown inputs.

According to Equations (50), (51), and (53), if $G_{dd}(s) \approx I_m$ can be appropriately designed and the functional observer gives a good estimate of the EID, such that $\hat{u}_f(t) \to u_f(t)$, then $\widetilde{d}_e(t) \to d_e(t)$ when $t \to \infty$ (see Equation (48)). Specifically, for any positive scalars $\varepsilon_1 << 1$, $\varepsilon_2 << 1$, $\varepsilon_3 << 1$ and assuming that the bounded $G_{ufo}(s) \neq 0$ has been properly determined, if

$$
1 - \left| G_{dd}^{ii}(s) \right| \le \varepsilon_1; \ (i = 1, \ldots, m)
\tag{54}
$$

and

$$
\sum_{j=1}^{m} \left| G_{dd}^{ij}(s) \right| \le \varepsilon_2; \ (i; j = 1, \ldots, m; \ j \neq i),
\tag{55}
$$

then from Equations (51) and (53), we can obtain

$$
\left\| G_{yd}^{ij}(s) \right\|_\infty \le \varepsilon_3; \ (i = 1, \ldots, p; j = 1, \ldots, m),
\tag{56}
$$

such that $y_c(t) \to 0$ when $t \to \infty$.

In other words, let $\Delta y_c(s)$ be defined as the error between the disturbed and undisturbed outputs, then $y_c(t) \to 0$ in Equation (53) implies that $\Delta y_c(s) = \left[ C(sI_n - A)^{-1}B + D \right]\left( d_e(s) - \hat{d}_e(s) \right)$, i.e., $\hat{d}_e(s) \approx d_e(s)$. According to Equations (37a) and (37b), it is true that $\widetilde{d}_e(s) \approx d_e(s)$ in steady state.

As previously mentioned, the design of $G_{dd}(s)$ should honor Equations (54) and (56), i.e., $G_{dd}(s) \approx I_m$, to effectively estimate the EID and cancel the unknown disturbance. Nevertheless, it is well-known that $\|G_{yd}(s)\|_\infty$ in Equation (56) cannot be ideally minimized throughout the frequency range $\omega_c \in [0, \infty)$. Therefore, the natural frequency of the unknown low-frequency disturbances to be suppressed is assumed to be below $\omega_c$, which is true for a wide class of servo systems. Thus, it is suggested to design $G_{dd}(s)$ as a low-pass-filter-type dynamic system, such that $\max\limits_{\omega \in \Omega} \|G_{yd}(j\omega)\|_\infty < 1$, where $\Omega \in [0, \omega_c)$ instead, to filter the higher frequency noise [24].

For simplicity, $G_{dd}(s)$ is directly constructed as the diagonal matrix

$$
G_{dd}(j\omega) = \mathrm{diag}\left\{ G_{dd}^{11}(j\omega_{11}), G_{dd}^{22}(j\omega_{22}), \ldots, G_{dd}^{mm}(j\omega_{mm}) \right\},
\tag{57}
$$

with the $1^{st}$-order filter (with least phase lag and simple structure)

$$
G_{dd}(j\omega) = \frac{1}{\frac{1}{\omega_{ii}}s + 1} = \frac{1}{\tau_{ii}s + 1} = \frac{\omega_{cii}}{s + \omega_{cii}},
\tag{58}
$$

where $\omega_{ii} \in \Omega_{ii}$, with $\Omega_{ii} \subseteq \Omega$, $i = 1, 2, \cdots, m$, $\omega_{cii}$ as the cut-off frequency of the $i^{th}$ diagonal element, and $\tau_{ii}$ as the corresponding time constant. Its state–space representation $(A_f, B_f, C_f)$ can be obtained from $G_{dd}(s)$ by an appropriate transformation. More details about selecting the low-pass filter from Equations (37a) and (37b) can be found in [24].

As in [4], the proposed approach preserves some independence in the design of the state observer, even though the dynamics of such an observer are associated with the EID estimator.

**Theorem 3.** *For a well-designed optimal tracker (Equation (40)), the control law (to be shown in Equation (75)) guarantees the stability of the control system under the following conditions: (i)$A - L_cC$ is stable; (ii) if matrix $\mathcal{L} \in \mathfrak{R}^{\kappa \times n}$ exists, such that Theorem 1 holds for the functional observer presented in Equations (13a) and (13b); and (iii) if the low-pass filter $G_{dd}(s) \approx I_m$, such that $\max\limits_{\omega \in \Omega}\|G_{yd}(j\omega)\|_{\infty} < 1$, where $\Omega \equiv [0, \omega_c)$.*

The proof of this theorem is similar to the proof of Theorem 4.4 in [4].

## 5. Design Procedure of the Unknown Input Linear Functional Observer for the Unknown Perturbed Singular Systems

Considering the unknown minimum-phase non-square continuous-time singular system with more control input channels than output channels, this section proposes a procedure for designing a high-performance optimal analog state estimate tracker and an EID estimator for the system with unknown mismatched input and output disturbances. First, we apply the off-line OKID method [17] to determine the parameters $(G_{ok}, H_{ok}, C_{ok}, D_{ok}, L_{ok})$ of the discrete-time mathematical model of this unknown system. Then, we carry out the discrete-to-continuous model conversion to obtain the continuous-time model that will be used to construct the analog unknown input functional observer to estimate and compensate the net EID. Lastly, we determine the proposed robust observer-based optimal LQAT for the unknown singular system. Three advanced techniques are now briefly described to achieve this goal.

**Part 1: Observer–Kalman-Filter Identification for the Unknown Singular System**

We consider the continuous-time impulsive-mode-free singular system (Equations (1a) and(1b)), i.e.,

$$E_r\dot{x}_c(t) = A_rx_c(t) + B_ru_c(t), \tag{59a}$$

$$y_c(t) = C_rx_c(t), \tag{59b}$$

where matrices $E_r$, $A_r$, $B_r$, and $C_r$ are assumed to be unknown. In theory, the above singular system can always be transformed into a corresponding low-order regular system with a direct transmission term from input to output

$$\dot{\hat{x}}_{c,s}(t) = \hat{A}_s\hat{x}_{c,s}(t) + \hat{B}_su_c(t), \tag{60a}$$

$$y_c(t) = \hat{C}_s\hat{x}_{c,s}(t) + \hat{D}_su_c(t). \tag{60b}$$

Here, it is assumed that $(\hat{A}_s, \hat{B}_s, \hat{C}_s, \hat{D}_s)$ are unknown system matrices of appropriate dimensions.

Step 1: Implement the off-line OKID method [17] to the singular system to determine the appropriate (low-) order of the singular system and the corresponding system matrices, with the corresponding sampling time $T_s$.

Step 2: Transform the discrete-time system–observer models obtained in Step 1 to the continuous-time system–observer models. Based on the identified discrete-time model $(G_{ok}, H_{ok}, C_{ok}, D_{ok})$, the corresponding continuous-time model $(A_{ok}, B_{ok}, C_{ok}, D_{ok})$ can be determined as

$$\dot{x}(t) = A_{ok}x(t) + B_{ok}u_c(t), \tag{61a}$$

$$y_c(t) = C_{ok}x(t) + D_{ok}u_c(t), \tag{61b}$$

where $x(t) \in \mathcal{R}^n$ is the state vector, $u_c(t) \in \mathcal{R}^m$ is the control input vector, $y_c(t) \in \mathcal{R}^p$ is the measurable output vector, $A_{ok} = \frac{1}{T_s} \ln G_{ok}$, $B_{ok} = A_{ok}(G_{ok} - I_n)^{-1} H_{ok}$, and $T_s$ is the sampling time.

**Part 2: Observer Design for Noisy Singular System**

For servo control, it is required to have more control inputs than outputs. However, for EID estimation, it is desirable to have more outputs than control inputs. Considering that the main purpose of this paper is to propose a robust tracker for a system subject to unknown disturbances, it is naturally assumed that the given system has more control inputs than outputs. To achieve this, the non-square system with $m > p$ is transformed into a square system with $m = p$ using an "artificial" coordinate [14], and finally transformed back to non-square and implemented in the original coordinate. These steps are now briefly shown.

Step 1: Assume that the corresponding regular system (Equations (60a) and (60b)) of order $n$, with $m$ inputs and $p$ outputs and subject to an unknown disturbance, is non-square ($m > p$), described by

$$\dot{\hat{x}}_{c,s}(t) = \hat{A}_s \hat{x}_{c,s}(t) + \hat{B}_s u_c(t) + d_i(t), \tag{62a}$$

$$y_c(t) = \hat{C}_s \hat{x}_{c,s}(t) + \hat{D}_s u_c(t) + d_o(t), \tag{62b}$$

where $\hat{A}_s \in \mathcal{R}^{n \times n}$, $\hat{B}_s \in \mathcal{R}^{n \times m}$, $\hat{C}_s \in \mathcal{R}^{p \times n}$, and $\hat{D}_s \in \mathcal{R}^{p \times m}$, are constant matrices. The mismatched input–output disturbances are given as $d_i(t) = G_i \bar{d}_i(t) \in \mathcal{R}^n$, $d_o(t) = G_o \bar{d}_o(t) \in \mathcal{R}^p$, respectively; with $l_i$ and $l_o$ representing the number of input and output disturbances, respectively; and $G_i \in \mathcal{R}^{n \times l_i}$, $G_o \in \mathcal{R}^{p \times l_o}$, $\bar{d}_i(t) \in \mathcal{R}^{l_i}$, and $\bar{d}_o(t) \in \mathcal{R}^{l_o}$ being unknown. Additionally, $l_i$ and $l_o$ may be greater than $m$ and $p$ and may even be no less than $n$. To fulfill the required assumptions for the linear functional observer (see Part 3), the transformation matrix $\eta \in \mathcal{R}^{m \times p}$ is determined, such that the system described in terms of $\left( \hat{A}_s, \hat{B}_a, \hat{C}_s, \hat{D}_a \right)$ for the control input $u_a(t)$

$$\dot{\hat{x}}_{c,s}(t) = \hat{A}_s \hat{x}_{c,s}(t) + \hat{B}_s \eta u_a(t) + d_i(t) = \hat{A}_s \hat{x}_{c,s}(t) + \hat{B}_a u_a(t) + d_i(t), \tag{63a}$$

$$y_c(t) = \hat{C}_s \hat{x}_{c,s}(t) + \hat{D}_s \eta u_a(t) + d_o(t) = \hat{C}_s \hat{x}_{c,s}(t) + \hat{D}_a u_a(t) + d_o(t), \tag{63b}$$

is square, controllable, and of the minimum phase, where $\hat{B}_a = \hat{B}_s \eta \in \mathcal{R}^{n \times p}$, $\hat{D}_a = \hat{D}_s \eta \in \mathcal{R}^{p \times p}$, $u_a(t) \in \mathcal{R}^p$, and

$$u_c(t) = \eta u_a(t) \in \mathcal{R}^m. \tag{64}$$

Step 2: Represent the system (Equations (63a) and (63b)) with mismatched input and output disturbances as a net EID, i.e.,

$$\dot{x}_e(t) = \hat{A}_s x_e(t) + \hat{B}_a [u_a(t) + d_{ea}(t)], \tag{65a}$$

$$y_c(t) = \hat{C}_s x_e(t) + \hat{D}_a [u_a(t) + d_{ea}(t)], \tag{65b}$$

where $d_{ea}(t) \in \mathcal{R}^p$ is the squared-down EID.

Step 3: Obtain the estimate $\hat{d}_{ea}(t)$ of $d_{ea}(t)$ in Part 3. To get $\widetilde{d}_{ea}(t)$, construct the low-pass filter

$$\dot{x}_f(t) = A_f x_f(t) + B_{fa} \hat{d}_{ea}(t), \tag{66a}$$

$$\widetilde{d}_{ea}(t) = C_f x_f(t), \tag{66b}$$

where $\hat{d}_{ea}(t)$ is the input, $\widetilde{d}_{ea}(t) \in \mathcal{R}^p$ is the output that estimates the equivalent unknown input, $x_e(t)$ is the state of the Luenberger observer [25], $A_f \in \mathcal{R}^{n_f \times n_f}$, $B_{fa} \in \mathcal{R}^{n_f \times p}$, and $C_f \in \mathcal{R}^{p \times n_f}$. In the frequency domain, the closed-loop gain $G_{yd}(s)$ from the disturbance $d_{ea}(t)$ to the output $y_c(t)$ cannot be minimized ideally through $\omega \in [0, \infty)$. For this reason, it is assumed that the natural frequencies of the unknown disturbances to be suppressed are below the cut-off frequency $\omega_c$.

Step 4: Construct the Luenberger observer

$$\dot{\hat{x}}_e(t) = A_{ok}\hat{x}_e(t) + B_{ok,a}u_{fa}(t) + L_c(y_c(t) - \hat{y}_c(t)), \tag{67a}$$

$$y_c(t) = C_{ok}\hat{x}_e(t) + D_{ok,a}u_{fa}(t), \tag{67b}$$

where $\hat{x}_e(t)$ and $\hat{y}_c(t)$ are the estimated state and output vectors, respectively. Here, $B_{ok,a} = B_{ok}\eta$, $D_{ok,a} = D_{ok}\eta$, and $L_c$ is the state observer gain. To further reduce the estimation errors, let $\overline{A} = A + hI_n$. Then, apply Lemma 1 with appropriate $(Q_o, R_o)$ to determine

$$L_c = P_o C_{ok}^T R_o^{-1}. \tag{68}$$

**Part 3: Design of the Linear Functional Observer**

We use the linear functional observer to estimate the $F\omega(t)$ in Equation (12). To achieve this goal based on the concept of EID and OKID methods, Equations (65a) and (65b) are represented as

$$\dot{x}_e(t) = A_{ok}x_e(t) + B_{ok,a}[u_a(t) + d_{ea}(t)], \tag{69a}$$

$$y_c(t) = C_{ok}x_e(t) + D_{ok,a}[u_a(t) + d_{ea}(t)]. \tag{69b}$$

Observe that $(A, B, C, D)$ and $u_f(t) \in \mathcal{R}^m$ in Equations (10a) and (10b) are now replaced by $(A_{ok}, B_{ok,a}, C_{ok}, D_{ok,a})$ and $u_{fa}(t) = [u_a(t) + d_{ea}(t)] \in \mathcal{R}^p$, respectively. Assumptions associated with the linear functional observer [11] include that $m \leq p$, and that the number of inputs of the equivalent system is equal to the number of outputs of the system after transforming it to the square in part 2. The procedure to construct the linear functional observer is now described in detail.

Step 1: Define the augmented state vector $\omega(t) = [\ x_e^T(t) \quad u_{fa}^T(t)\ ]^T \in \mathcal{R}^{(n+p)}$, such that the system (Equation (69a) and (69b)) can be expressed as

$$\overline{E}_\omega \dot{\omega}(t) = \overline{A}_\omega \omega(t), \tag{70a}$$

$$y_c(t) = \overline{C}_\omega \omega(t), \tag{70b}$$

where $\overline{E}_\omega = \begin{bmatrix} I_{n\times n} & 0_{n\times p} \\ 0_{p\times n} & 0_{p\times p} \end{bmatrix} \in \mathcal{R}^{(n+p)\times(n+p)}$, $\overline{A}_\omega = \begin{bmatrix} A_{ok} & B_{ok,a} \\ 0_{p\times n} & 0_{p\times p} \end{bmatrix} \in \mathcal{R}^{(n+p)\times(n+p)}$, and $\overline{C}_\omega = [\ C_{ok} \quad D_{ok,a}] \in \mathcal{R}^{p\times(n+p)}$. Define the functional state vector $z(t) \in \mathcal{R}^\kappa$ that must be reconstructed (or estimated) as

$$z(t) = F\omega(t) = \begin{bmatrix} F_1 & 0 \\ 0 & I_p \end{bmatrix}\begin{bmatrix} x_e(t) \\ u_{fa}(t) \end{bmatrix} = \begin{bmatrix} F_1 x_e(t) \\ u_{fa}(t) \end{bmatrix}, \tag{71}$$

where $F_1 \in \mathcal{R}^{(\kappa-p)\times n}$, and $F \in \mathcal{R}^{\kappa\times(n+p)}$ is a given constant matrix, which satisfies $rank(F) = \kappa$ and $rank\begin{bmatrix} \overline{C}_\omega \\ F \end{bmatrix} = (p + \kappa) \leq (n + p)$. To estimate $z(t)$, we construct the reduced-order functional observer of order $\kappa$ for the system (Equations (69a) and (69b))

$$\dot{w}(t) = Nw(t) + Jy_c(t), \tag{72a}$$

$$\hat{z}(t) = w(t) + Qy_c(t). \tag{72b}$$

Therefore, the estimated unknown input vector becomes

$$\hat{u}_{fa}(t) = [\ 0_{p\times(\kappa-p)} \quad I_p\ ]\hat{z}(t) = [\ 0_{p\times(\kappa-p)} \quad I_p\ ]\begin{bmatrix} F_1\hat{x}_e(t) \\ \hat{u}_{fa}(t) \end{bmatrix} \in \mathcal{R}^p. \tag{73}$$

Thus, the estimated EID $\hat{d}_{ea}(t) = \hat{u}_{fa}(t) - u_a(t)$ can be derived.

Step 2: Check the existence of Condition 1 of Theorem 2 to see whether

$$
rank\begin{bmatrix} F\overline{A}_\omega & F \\ \overline{C}_\omega\overline{A}_\omega & \overline{C}_\omega \\ \overline{C}_\omega & 0 \\ 0 & \overline{E}_\omega \\ F & 0 \end{bmatrix} = rank\begin{bmatrix} \overline{C}_\omega\overline{A}_\omega & \overline{C}_\omega \\ \overline{C}_\omega & 0 \\ 0 & \overline{E}_\omega \\ F & 0 \end{bmatrix}
\tag{74a}
$$

is satisfied or not. If not, repeat Step 1 by choosing another $F$ in terms of $F_1$ in Equation (71).

Step 3: Systematically derive all observer parameters.

(i)　Check the existence of Condition 2 of Theorem 2 to see whether

$$
rank\begin{bmatrix} sF - F\overline{A}_\omega & -F \\ \overline{C}_\omega\overline{A}_\omega & \overline{C}_\omega \\ \overline{C}_\omega & 0 \\ 0 & \overline{E}_\omega \end{bmatrix} = rank\begin{bmatrix} \overline{C}_\omega\overline{A}_\omega & \overline{C}_\omega \\ \overline{C}_\omega & 0 \\ 0 & \overline{E}_\omega \\ F & 0 \end{bmatrix} \quad \forall s \in C,\ \Re(s) \geq 0
\tag{74b}
$$

is satisfied or not. If yes, use Equation (34) to obtain $\Phi$ and $\Omega$. If not, which implies the pair $(\Phi, \Omega)$ is not detectable, stop, since a stable observer does not exist;

(ii)　Using Equation (33), derive $Z$, such that the matrix $N$ is stable;

(iii)　From Equation (32), obtain matrices $Q$, $T$, and $\mathcal{L}$. Substitute $Q$, $T$, and $\mathcal{L}$ into Equation (27) and check the necessary and sufficient condition to ensure that the matrix $N$ is a Hurwitz function.

(iv)　Finally, obtain matrix $J$ from Equation (27), i.e., $J = T + NQ$. As a result, the unknown matrices $N$, $\mathcal{L}$, $J$, $Q$, are determined and the observer design is, thus, completed.

Step 4: Design the optimal linear quadratic tracker with an appropriate weighting matrix pair $Q_c \gg R_c$ to determine the OKID-based optimal control law [18] as

$$
u_{fa}(t) = -K_c\hat{x}_e(t) + E_c r(t),
\tag{75}
$$

where $K_c = \overline{R}_c^{-1}\left(B_{ok,a}^T P + N_c^T\right)$, $E_c = -\overline{R}_c^{-1}\left[\left(C_{ok} - D_{ok,a}K_c\right)\left(A_{ok} - B_{ok,a}K_c\right)^{-1}B_{ok,a} + D_{ok,a}\right]^T Q_c$, in which $\overline{R}_c = R_c + D_{ok,a}^T Q_c D_{ok,a}$, and $N_c = C_{ok,a}^T Q_c D_{ok,a}$, and $P > 0$ satisfies the algebraic Riccati equation

$$
A_{ok}^T P + PA_{ok} - \left(PB_{ok,a} + N_c\right)\overline{R}_c^{-1}\left(B_{ok,a}^T P + N_c^T\right) + C_{ok}^T Q_c C_{ok} = 0.
\tag{76}
$$

Then, implement the controller

$$
u_c(t) = \eta u_a(t) = \eta\left(u_{fa}(t) - \widetilde{d}_{ea}(t)\right).
\tag{77}
$$

The architecture of the compensation improvement OKID-based LQAT disturbance estimator with a low-pass filter for the singular system is depicted in Figure 1. If the corresponding slow subsystem of the given singular system $(A_{ok}, B_{ok}, C_{ok}, D_{ok})$ is square, then set $\eta = I_m$ in Equation (77) and Figure 1.

Remark 1 Consider a class of linear strictly proper regular systems with an uncertain time-varying state delay and an unknown input vector (to be estimated). The system also has another unknown disturbance vector, which might contain either the parameter uncertainties or non-linearities, and can be treated as an additional unknown input vector to the system. As a result, two unknown input vectors exist that act on the system. The method developed in [11] consists of designing reduced-order functional observers to simultaneously estimate the states and the first unknown input vector (but not the unknown disturbance vector, which represents uncertainties or non-linearities) of the aforementioned systems (see Chapter 7 in [11]). As an alternative, in the approach proposed in this paper, the uncertainties or state-dependent non-linear disturbances have been theoretically merged into the first unknown input

vector, such that the combined unknown input vector is estimated. From the point of view of the design methodology, it is important to point out that the proposed two-norm minimization approach performs well for the system addressed in this paper, however not for general uncertain systems or non-linear systems. Specifically, the H-infinity-norm minimization approach is suggested for uncertain systems and will be considered as a future research topic.

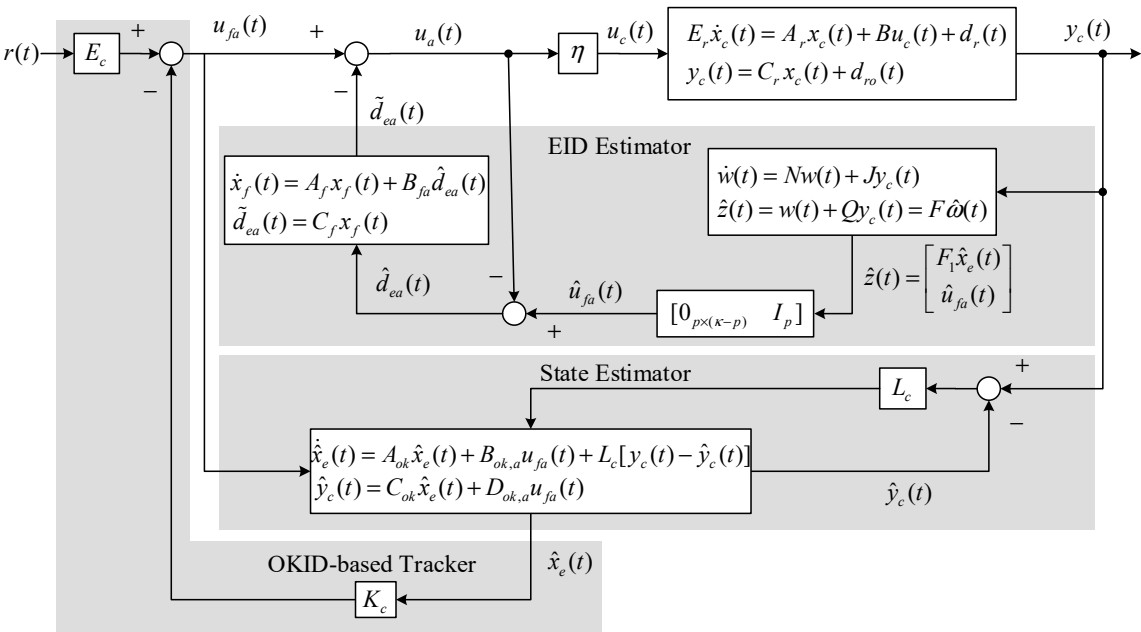

**Figure 1.** An observer–Kalman-filter (OKID)-based linear quadratic analog tracker (LQAT) with a functional observer for non-square singular systems. EID, equivalent input disturbance.

## 6. Illustrative Examples

This section presents two examples to illustrate the application of the proposed approach and to show that it outperforms the traditional approach.

**Example 1.** *Square MIMO System*

Consider the continuous-time singular system without the impulsive mode described as

$$E_r \dot{x}(t) = A_r x(t) + B_r u_c(t) + d_r(t),  \tag{78a}$$

$$y_c(t) = C_r x(t) + d_{ro}(t),  \tag{78b}$$

where

$$E_r = \begin{bmatrix} 1 & 0 & 0 & 0 & 0 & 0 \\ 0 & 1 & 0 & 0 & 0 & 0 \\ 0 & 0 & 1 & 0 & 0 & 0 \\ 1 & 0 & -1 & 0 & 0 & 1 \\ 0 & 1 & 0 & 0 & 0 & 0 \\ 0 & 0 & 0 & 0 & 0 & 1 \end{bmatrix}, A_r = \begin{bmatrix} 7 & 6 & 5 & 6 & -9 & 5 \\ -2 & 1 & -1 & 3 & 0 & 4 \\ -10 & -7 & -6 & -4 & 10 & 4 \\ -4 & 1 & -7 & -7 & 8 & -4 \\ 7 & -4 & -5 & -5 & 2 & 2 \\ -4 & 4 & 3 & 4 & 5 & 2 \end{bmatrix},$$

$$B_r = \begin{bmatrix} 6 & 4 \\ -10 & 6 \\ -6 & -4 \\ 10 & 3 \\ 6 & -9 \\ 3 & 1 \end{bmatrix}, \quad C_r = \begin{bmatrix} 1 & -6 & 10 & 2 & -1 & -3 \\ 7 & 9 & -3 & 1 & -3 & -8 \end{bmatrix},$$

$$x(0) = \begin{bmatrix} 0.5 & 0.05 & 0 & 0.7862 & 1.2448 & 0.225 \end{bmatrix}^T.$$

The singular system can be transformed into the equivalent regular system with an input–output direct feedthrough term

$$\dot{x}_s(t) = \hat{A}_s x_s(t) + \hat{B}_s u_c(t) + d_s(t), \tag{79a}$$

$$y_c(t) = \hat{C}_s x_s(t) + \hat{D}_s u_c(t) + d_{so}(t), \tag{79b}$$

where $x_s(0) = \begin{bmatrix} 0.5 & 0.05 & 0 & 0.225 \end{bmatrix}^T$,

$$\hat{A}_s = \begin{bmatrix} -4.4403 & 0.9627 & -2.7015 & 2.7164 \\ 3.1940 & -0.7463 & -2.0299 & 3.3284 \\ 7.3284 & -2.9552 & 1.6418 & 5.9403 \\ 15.0522 & 2.5299 & 4.7612 & 1.6269 \end{bmatrix}, \quad \hat{B}_s = \begin{bmatrix} -3.0448 & 7.6940 \\ -1.8955 & -1.1194 \\ 11.2537 & -14.4328 \\ 27.8358 & -18.4552 \end{bmatrix},$$

$$\hat{C}_s = \begin{bmatrix} 2.0373 & -7.3358 & 8.6866 & -3.5522 \\ 1.4552 & 7.9030 & -5.2239 & -8.5373 \end{bmatrix}, \quad \hat{D}_s = \begin{bmatrix} 2.5970 & -2.7537 \\ -5.7164 & 3.6045 \end{bmatrix},$$

with

$$d_s(t) = G_i \bar{d}_s(t) = \begin{bmatrix} -1 & 2 & 5 \\ 4 & -5 & 3 \\ 3 & 3 & 4 \\ 5 & 4 & -1 \end{bmatrix} \bar{d}_s(t),$$

where

$$\bar{d}_s(t) = \begin{bmatrix} 0.1(20\sin(4\pi t) + \cos(5\pi t) + \sin(\pi t) + \sin(5\pi t) - 1) + \sin(x_{s1}(t) + x_{s3}(t)) \\ 0.1(20\sin(5\pi t) + \cos(2\pi t) + \sin(\pi t) + \sin(2\pi t) - 1) + \cos(x_{s2}(t)) \\ 0.1(20\sin(3\pi t) + \cos(4\pi t) + \sin(3\pi t) + \sin(2\pi t) - 1) + \sin(x_{s4}(t)) \end{bmatrix} \in \mathfrak{R}^{l_i}$$

for $t = 10 \sim 30$ sec, or $\bar{d}_s(t) = 0_{l_i \times 1}$ otherwise

$$d_{so}(t) = G_o \bar{d}_{so}(t) = \begin{bmatrix} -4 & 7 \\ 3 & -1 \end{bmatrix} \bar{d}_{so}(t),$$

where $\bar{d}_{so}(t) = \begin{bmatrix} 0.1(\sin(\pi t) + \sin(5\pi t) - 1) \\ 0.1(\sin(2\pi t) - 1) \end{bmatrix} \in \mathfrak{R}^{l_o}$, for $t = 10 \sim 30$ sec, or $\bar{d}_{so}(t) = 0_{l_o \times 1}$ otherwise. It is assumed that $G_i$, $G_o$, $\bar{d}_s(t)$ and $\bar{d}_{so}(t)$ are unknown. Additionally, there are more disturbances than outputs. The eigenvalues and zeros of the open-loop singular system are $\{-9.1996, -3.4145, 3.6372, 7.0590\}$ and $\{-33.4278 \pm 23.8397i, -2.0359 \pm 2.0429i\}$, respectively. The desired output trajectory is given by $r(t) = [r_1(t), r_2(t)]^T$, where

$$r_1(t) = \begin{cases} 0 & , 9 < t \text{ sec} \\ \sin(0.1\pi t) + \cos(0.2\pi t) - \sin(0.5\pi t) \times \cos(0.3\pi t) & , 9 \leq t < 30 \text{ sec} \\ -0.5 & , t \geq 30 \text{ sec} \end{cases},$$

$$r_2(t) = \begin{cases} 1 & , 9 < t \text{ sec} \\ \sin(0.2\pi t) + \cos(0.1\pi t) + \sin(0.5\pi t) \times \cos(0.5\pi t) & , 9 \leq t < 30 \text{ sec} \\ 2 & , t \geq 30 \text{ sec} \end{cases}.$$

Figure 2 shows the significant time-varying features of $d_{s1}(t)$ without state-dependent disturbance (bounded by $\pm 13$) and $d_{so1}(t)$ (bounded by $\pm 2$) when compared with the smooth desired output trajectory $r(t)$ (bounded by $\pm 2$), which makes this example more challenging.

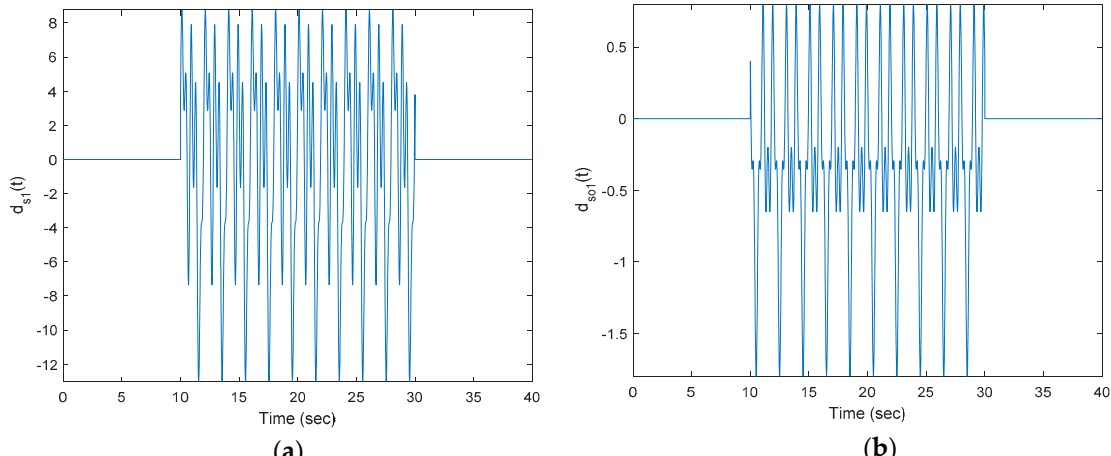

**Figure 2.** (**a**) Input disturbance $d_{s1}(t)$ (without state-dependent disturbance); (**b**) output disturbance $d_{so1}(t)$.

Now, assume that the singular system is unknown. The objective of this paper is to design a robust optimal OKID-based LQAT with a modified functional observer-based EID estimator, such that the controlled square system has an improved tracking performance.

Step 1: Identification of the singular system.

It is suitable to assume that the unknown disturbances occur during some unexpected periods, such that a set of disturbance-free input–output data of the open-loop system can be collected and an appropriate-order (4 this example) open-loop model can be identified by applying the off-line OKID method. As a consequence, the converted identified system matrices in the continuous-time domain are given as

$$A_{ok} = \begin{bmatrix} 224.4093 & 20.3493 & -203.0721 & -1.1996 \\ -20.2149 & 189.8678 & 0.7314 & -212.3123 \\ 203.0742 & -1.3015 & -181.2974 & 20.4771 \\ 0.4529 & 212.3102 & -20.5253 & -234.8976 \end{bmatrix}, B_{ok} = \begin{bmatrix} -36.393 & 29.3443 \\ 26.0444 & -15.3636 \\ -36.9952 & 30.6218 \\ 26.8896 & -16.8004 \end{bmatrix},$$

$$C_{ok} = \begin{bmatrix} -2.1609 & -2.862 & -1.9911 & -2.6455 \\ 2.9487 & -2.1395 & 2.7195 & -1.9425 \end{bmatrix}, D_{ok} = \begin{bmatrix} 2.5973 & -2.7555 \\ -5.7322 & 3.6162 \end{bmatrix}.$$

The eigenvalues and zeros of the open-loop OKID model are $\{-9.1909, -3.4346,\ 3.7011,\ 7.0066\}$ and $\{-33.3957 \pm 23.7553i,\ -2.0351 \pm 1.8661i\}$, respectively.

Step 2: Determination of the state observer.

Error equations are constructed by choosing the weighting matrix pairs $\{ Q_o,\ R_o \} = \left\{ 10^4 I_4,\ I_2 \right\}$ for the determination of $L_c$ with $h = 50$. This results in

$$L_c = \begin{bmatrix} -115.3191 & 156.8902 \\ -138.9246 & -103.4300 \\ -24.5142 & 32.8421 \\ -35.1095 & -25.3732 \end{bmatrix}.$$

The filter matrices $\left(A_f, B_{fa}, C_f\right)$ are chosen as $A_f = \begin{bmatrix} -100 & 0 \\ 0 & -100 \end{bmatrix}$, $B_{fa} = \begin{bmatrix} 8 & 0 \\ 0 & 8 \end{bmatrix}$, and $C_f = \begin{bmatrix} 12.5 & 0 \\ 0 & 12.5 \end{bmatrix}$, so that $C_f(sI - A_f)^{-1}B_{fa} \approx I_m$ for $s = j\omega = 0$. Note that the selected eigenvalues of $A_f$ are sufficiently negative to filter out high-frequency disturbances.

Step 3: Design of the functional observer according to Section 4 to estimate the linear combination of state and unknown input $F\omega(t)$ in Equation (71).

Choose $F = \begin{bmatrix} 1 & 0 & 0 & 0 & 0 & 0 \\ 0 & 1 & 0 & 0 & 0 & 0 \\ 0 & 0 & 0 & 0 & 1 & 0 \\ 0 & 0 & 0 & 0 & 0 & 1 \end{bmatrix}$, such that the ranks of those matrices checked in Theorem 2

are all equal to 12 and the unknown input $\hat{u}_{fa}(t)$ can be estimated. According to Section 4, the condition given in this example implies that all theorems hold, thus the linear functional observer

$$\dot{w}(t) = Nw(t) + Jy_c(t), w(0) = 0_{4\times1},\tag{80a}$$

$$\dot{w}(t) = Nw(t) + Jy_c(t), \hat{z}(t) = w(t) + Qy_c(t) \approx F\omega(t),\tag{80b}$$

is obtained, where

$$\mathcal{L} = \begin{bmatrix} 1.0000 & 0.0000 & 0.0000 & 0.0000 & 0 & 0 \\ 0.0000 & 1.0000 & 0.0000 & 0.0000 & 0 & 0 \\ 0.0486 & -2.5373 & 0.0458 & -2.3302 & 0 & 0 \\ -0.7384 & -3.4302 & -0.6794 & -3.1565 & 0 & 0 \end{bmatrix} \in \mathcal{R}^{\kappa\times(n+m)}$$

(mentioned in Theorem 1),

$$N = \begin{bmatrix} 444.673 & 19.807 & -406.926 & 303.26 \\ -20.905 & 421.012 & 110.864 & -10.718 \\ 39.759 & -2158.599 & -508.808 & 14.632 \\ -600.263 & -2918.42 & -82.242 & -427.738 \end{bmatrix}$$

is a stable matrix,

$$J = \begin{bmatrix} -5.7167 & 3.7585 \\ -0.9552 & -4.9763 \\ 2.1877 & 24.0545 \\ 12.389 & 26.9334 \end{bmatrix},$$

and

$$Q = \begin{bmatrix} 0 & 0 \\ 0 & 0 \\ -0.5648 & -0.4304 \\ -0.8953 & -0.4057 \end{bmatrix}.$$

Step 4: Design of the linear-quadratic state feedback tracker integrated with the EID estimator.

In the LQAT design, the weighting matrix pair $\{Q_c, R_c\} = \left\{10^4 I_2, I_2\right\}$ is chosen to obtain the optimal control law $u_c(t) = -K_c\hat{x}_e(t) + E_cr(t) - \tilde{d}_e(t)$, where

$$K_c = \begin{bmatrix} -0.0501 & 2.5368 & -0.0444 & 2.3302 \\ 0.7365 & 3.4298 & 0.6812 & 3.1562 \end{bmatrix} \text{ and } E_c = \begin{bmatrix} -0.5648 & -0.4303 \\ -0.8952 & -0.4056 \end{bmatrix}.$$

Step 5: Simulation results.

First, to show the superiority of the proposed tracker, Figure 3 plots the input disturbance $d_{s1}(t)$ (state-dependent disturbance inside) and the pure non-linear state-dependent disturbance. All of

these disturbances (shown by parts) indicate a highly time-varying phenomenon. Figure 4a shows the time response of the controlled system and Figure 4b shows the output tracking errors under the control action of the traditional optimal OKID-based tracker without an EID estimator.

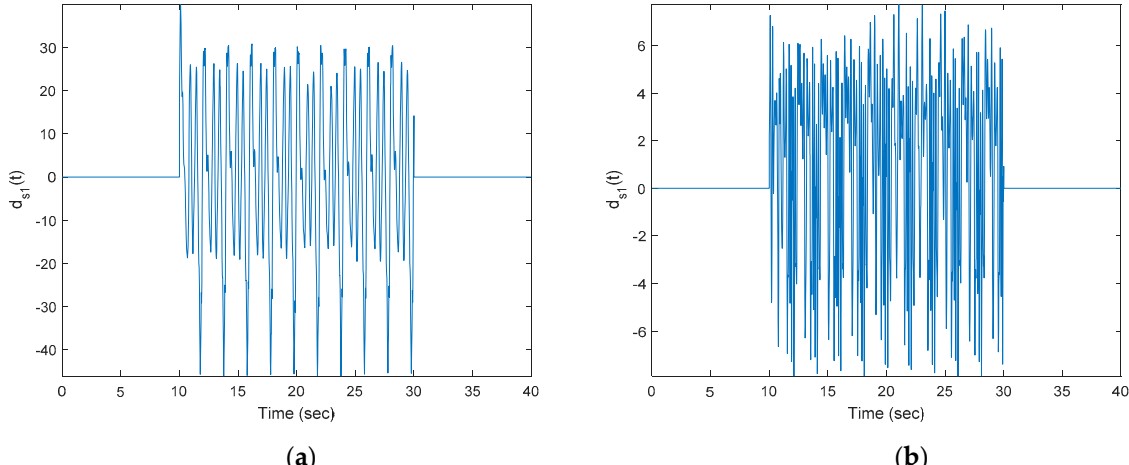

**Figure 3.** (**a**) Input disturbance $d_{s1}(t)$ (state-dependent disturbance inside) and (**b**) state-dependent disturbance in $d_{s1}(t)$.

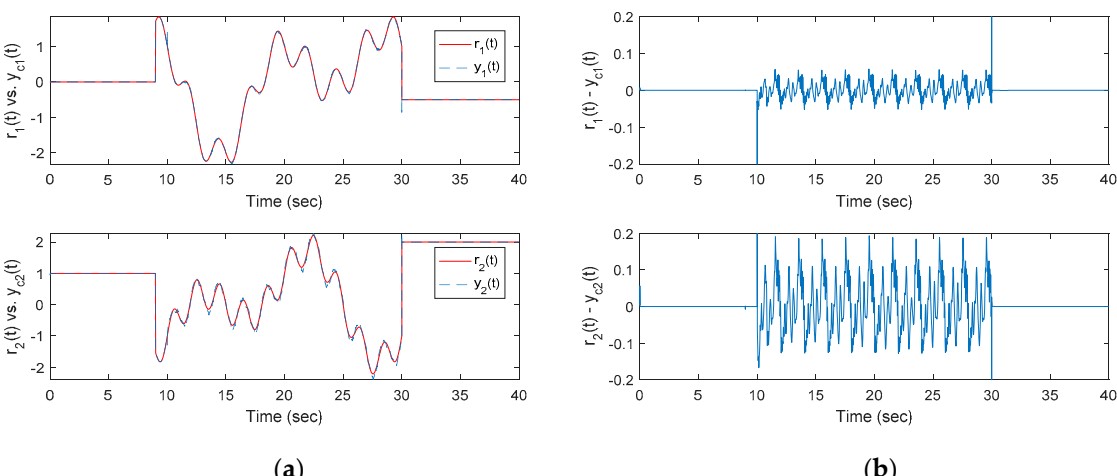

**Figure 4.** (**a**) Time responses of the closed-loop noisy square singular system for the optimal OKID-based LQAT without the EID estimator, (**b**) tracking errors of output $e(t) = r(t) - y_c(t)$.

Second, Figure 5 shows the input disturbance $d_{s1}(t)$ (state-dependent disturbance inside) and the pure non-linear state-dependent disturbance. Here, $d_{s1}(t)$ is slightly different from the optimal OKID-based LQAT without the EID estimator, because the differences between the two states $x_s(t)$ are induced by different controllers. Next, the robustness of the optimal OKID-based LQAT integrated with the functional observer-based EID estimator is shown in Figure 6a,b. Also, Figure 7 plots the control inputs corresponding to the proposed approach. Notice that $u_c(t) = u_a(t)$ for this square system, where $\eta = I_p$. A comparison with Figure 4 shows that the proposed approach (Figure 6) significantly improves the tracking errors of the noisy singular system.

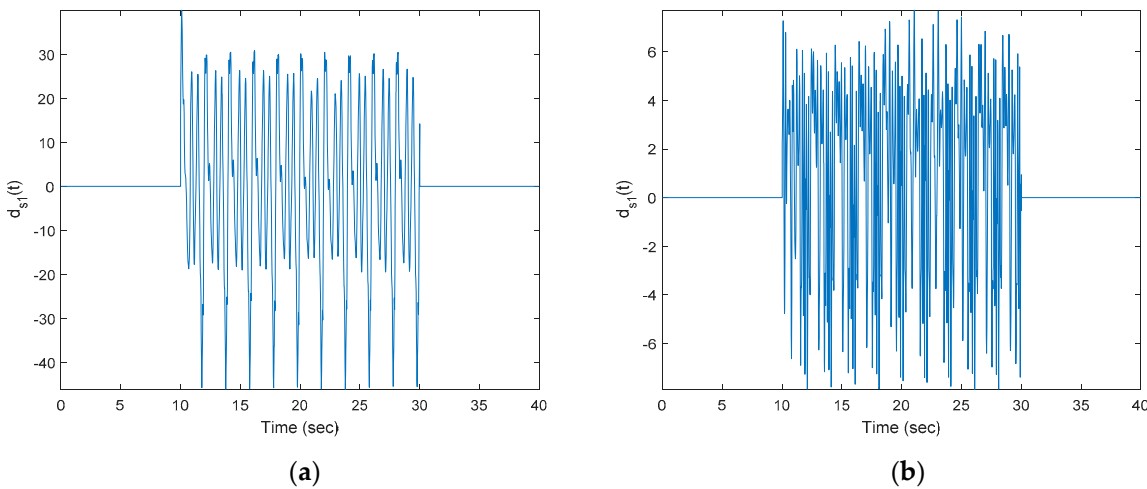

(**a**)                    (**b**)

**Figure 5.** (**a**) Input disturbance $d_{s1}(t)$ (state-dependent disturbance inside) and (**b**) state-dependent disturbance in $d_{s1}(t)$.

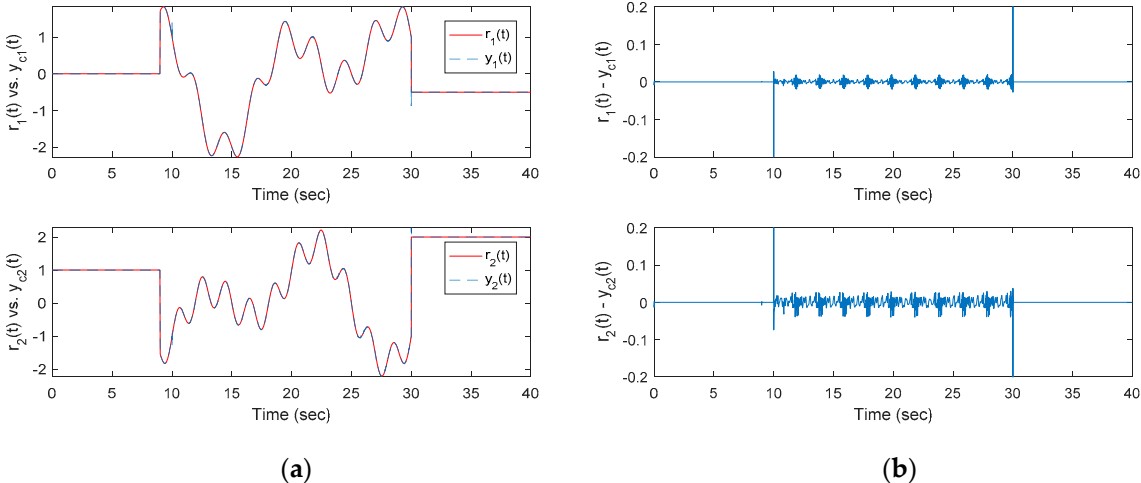

(**a**)                    (**b**)

**Figure 6.** (**a**) Time responses of the proposed method and (**b**) errors of output $e(t) = r(t) - y_c(t)$.

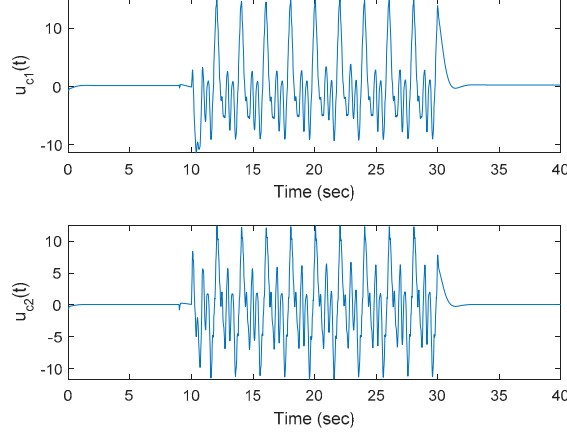

**Figure 7.** Control inputs of the OKID-based LQAT tracker with the proposed EID compensation for the square system with mismatched input and output disturbances.

**Example 2.** *Non-Square MIMO System*

Consider the continuous-time non-square singular system without impulsive mode described by

$$E_r \dot{x}(t) = A_r x(t) + B_r u_c(t) + d_r(t), \tag{81a}$$

$$y_c(t) = C_r x(t) + d_{ro}(t), \tag{81b}$$

where $E_r$ and $A_r$ are given in Example 1,

$$B_r = \begin{bmatrix} 8 & -9 & -4 \\ 2 & -1 & 3 \\ 4 & 3 & 8 \\ 5 & -4 & 7 \\ -2 & 0 & -6 \\ -5 & -5 & -2 \end{bmatrix}, C_r = \begin{bmatrix} 8 & 9 & -7 & 3 & -5 & -5 \\ -5 & 2 & 7 & 1 & -9 & -3 \end{bmatrix},$$

$$x(0) = \begin{bmatrix} 0.15 & 0.015 & 0 & 0.2661 & 0.3593 & -0.0675 \end{bmatrix}^T.$$

The singular system can be converted into the equivalent regular system with an input–output direct feedthrough term

$$\dot{x}_s(t) = \hat{A}_s x_s(t) + \hat{B}_s u_c(t) + d_s(t),$$
$$y_c(t) = \hat{C}_s x_s(t) + \hat{D}_s u_c(t) + d_{so}(t), \ x_s(0) = \begin{bmatrix} 0.15 & 0.015 & 0 & -0.0675 \end{bmatrix}^T,$$

where

$$\hat{A}_s = \begin{bmatrix} -4.4403 & 0.9627 & -2.7015 & 2.7164 \\ 3.1940 & -0.7463 & -2.0299 & 3.3284 \\ 7.3284 & -2.9552 & 1.6418 & 5.9403 \\ 15.0522 & 2.5299 & 4.7612 & 1.6269 \end{bmatrix}, \hat{B}_s = \begin{bmatrix} 12.3881 & -3.6045 & 9.2313 \\ -0.2388 & -1.0896 & -2.3731 \\ -2.8657 & -3.0746 & -11.4776 \\ -12.9104 & -8.2164 & -22.4851 \end{bmatrix},$$

$$\hat{C}_s = \begin{bmatrix} 1.0672 & 6.3955 & -11.1642 & -6.194 \\ -25.097 & -0.1269 & 1.0149 & -4.1642 \end{bmatrix}, \hat{D}_s = \begin{bmatrix} 2.6866 & 3.0075 & 7.9478 \\ 8.1194 & 5.5448 & 22.1866 \end{bmatrix},$$

with the same mismatched input disturbances (but without state-dependent disturbances) and output disturbances $d_{so}(t)$ described in Example 1,

$$d_s(t) = G_i \bar{d}_s(t) = \begin{bmatrix} -1 & 2 & 5 \\ 4 & -5 & 3 \\ 3 & 3 & 4 \\ 5 & 4 & -1 \end{bmatrix} \bar{d}_s(t),$$

$$\bar{d}_s(t) = \begin{bmatrix} 0.1(20\sin(4\pi t) + \cos(5\pi t) + \sin(\pi t) + \sin(5\pi t) - 1) \\ 0.1(20\sin(5\pi t) + \cos(2\pi t) + \sin(\pi t) + \sin(2\pi t) - 1) \\ 0.1(20\sin(3\pi t) + \cos(4\pi t) + \sin(3\pi t) + \sin(2\pi t) - 1) \end{bmatrix} \in \mathbb{R}^{l_1},$$

for $t = 10 \sim 30$ sec, or $\bar{d}_s(t) = 0_{l_1 \times 1}$ otherwise.

Similarly, it is assumed that $G_i$, $G_o$, $\bar{d}_s(t)$ and $\bar{d}_{so}(t)$ are unknown. Eigenvalues and "control zeros" [26] of the open-loop singular system are $\{-9.1996, -3.4145, 3.6372, 7.0590\}$ and $\{-3.4606 \times 10^5, -30.9692, -1.5141 \pm 1.8765i\}$, respectively. Also, the desired output trajectory $r(t)$ is the same as in Example 1.

Now, assume that the singular system is unknown. The objective is to design a robust optimal OKID-based LQAT with a modified functional observer-based EID estimator, such that the tracking performance of the controlled non-square system is improved.

Step 1: Identification of the singular system.

Apply the off-line OKID method to determine the appropriate order as 4, and the converted identified system matrices in the continuous-time domain are given as

$$
A_{ok} = \begin{bmatrix} 212.1358 & -2.6065 & -206.2029 & 1.861 \\ 2.526 & 202.0124 & -1.9511 & -208.864 \\ 206.2035 & 1.8947 & -200.1689 & -2.7866 \\ -1.7579 & 208.8631 & 2.8165 & -215.8972 \end{bmatrix}, \; B_{ok} = \begin{bmatrix} 10.5226 & -19.5544 & -14.9769 \\ 40.2286 & -6.32 & 41.3521 \\ 10.7314 & -19.3488 & -14.6376 \\ 40.2016 & -6.5768 & 40.5922 \end{bmatrix},
$$

$$
C_{ok} = \begin{bmatrix} -2.7742 & 2.3193 & -2.5354 & 2.1565 \\ -2.3362 & -2.7567 & -2.1735 & -2.5158 \end{bmatrix}, \; D_{ok} = \begin{bmatrix} 2.6927 & 3.0112 & 7.9609 \\ 8.1064 & 5.5509 & 22.1791 \end{bmatrix}.
$$

Eigenvalues and "control zeros" of the open-loop OKID model are $\{-9.2013, -3.3976, 3.5768, 7.1041\}$ and $\{-3.4777 \times 10^5, -30.9407, -1.512 \pm 1.8739i\}$, respectively. Appropriately choose a transformation matrix $\eta$ such that the system $(A_{ok}, B_{ok}\eta, C_{ok}, D_{ok}\eta)$ is square and of the minimum phase. Here, $\eta$ is chosen as $\eta = \begin{bmatrix} -8 & -9 \\ 6 & 1 \\ 3 & 4 \end{bmatrix}$, while eigenvalues and "control zeros" of the transformed system are $\{-9.2013, -3.3976, 3.5768, 7.1041\}$ and $\{-265.2835, -15.646, -1.1069 \pm 0.9538i\}$, respectively. Therefore, $B_{ok}$ and $D_{ok}$ are replaced by

$$
B_{ok,a} = \eta B_{ok} = \begin{bmatrix} -246.4379 & -174.1652 \\ -235.6925 & -202.9687 \\ -245.8564 & -174.4816 \\ -239.2965 & -206.022 \end{bmatrix}
$$

and $D_{ok,a} = \eta D_{ok} = \begin{bmatrix} 20.4077 & 10.6199 \\ 34.9912 & 21.3096 \end{bmatrix}$, respectively.

Step 2: Determination of the state observer.

Let the low-pass filter $\left(A_f, B_f, C_f\right)$ be the same as in Example 1. Also, based on the same parameters specified in Step 2 of Example 1

$$
L_c = \begin{bmatrix} -142.854 & 120.2801 \\ 11.1893 & -138.0508 \\ -31.9198 & -27.344 \\ 27.7948 & -32.4321 \end{bmatrix}.
$$

Step 3: Design the modified functional observer according to Section 4 to estimate the linear combination of the state and unknown input $F\omega(t)$ in Equation (71).

Choose $F = \begin{bmatrix} 1 & 0 & 0 & 0 & 0 & 0 \\ 0 & 1 & 0 & 0 & 0 & 0 \\ 0 & 0 & 0 & 0 & 1 & 0 \\ 0 & 0 & 0 & 0 & 0 & 1 \end{bmatrix}$, such that the ranks of those matrices to be checked in Theorem 2 are all equal to 12 and the unknown input $\hat{u}_{fa}(t)$ can be estimated. According to Section 4, all theorems hold due to the condition given in this example, and thus in the linear functional observer

$$
\dot{w}(t) = Nw(t) + Jy_c(t), \tag{82a}
$$

$$
\hat{z}(t) = w(t) + Qy_c(t) \approx F\omega(t), \tag{82b}
$$

where

$$
\mathcal{L} = \begin{bmatrix} 1.0000 & 0.0000 & 0.0000 & 0.0000 & 0 & 0 \\ 0.0000 & 1.0000 & 0.0000 & 0.0000 & 0 & 0 \\ 0.5422 & -1.2437 & 0.4890 & -1.1485 & 0 & 0 \\ -0.7806 & 2.1716 & -0.7010 & 2.0039 & 0 & 0 \end{bmatrix} \in \mathfrak{R}^{\kappa \times (n+m)}
$$

(mentioned in Theorem 1),

$$
N = \begin{bmatrix} 436.085 & -2.581 & -2601.806 & -1523.144 \\ 2.633 & 429.101 & -1095.315 & -799.863 \\ 448.212 & -1041.115 & -64.008 & 345.772 \\ -645.445 & 1817.85 & -708.615 & -1084.321 \end{bmatrix}
$$

is a stable matrix,

$$
J = \begin{bmatrix} 13.319 & -14.811 \\ 32.865 & -25.904 \\ -65.250 & 47.033 \\ 118.188 & -86.838 \end{bmatrix},
$$

and

$$
Q = \begin{bmatrix} 0 & 0 \\ 0 & 0 \\ 0.3368 & -0.1678 \\ -0.553 & 0.3225 \end{bmatrix}.
$$

Step 4: Design the linear-quadratic state feedback tracker integrated with the EID estimator.

In the LQAT design, the weighting matrix pair $\{Q_c, R_c\} = \{10^4 I_2, I_2\}$ is chosen to obtain the optimal control law $u_a(t) = -K_c \hat{x}_e(t) + E_c r(t) - \tilde{d}_e(t)$, where

$$
K_c = \begin{bmatrix} -0.5422 & 1.2438 & -0.489 & 1.1484 \\ 0.7807 & -2.1717 & 0.701 & -2.0037 \end{bmatrix} \text{ and } E_c = \begin{bmatrix} 0.3368 & -0.1678 \\ -0.553 & 0.3225 \end{bmatrix}.
$$

Step 5: Simulation results.

First, Figure 8a plots the time responses of the non-square system and Figure 8b shows the output tracking errors under the control action of the traditional optimal OKID-based tracker without an EID estimator. It can be concluded that the proposed method gives significantly improved performance compared to the traditional approach.

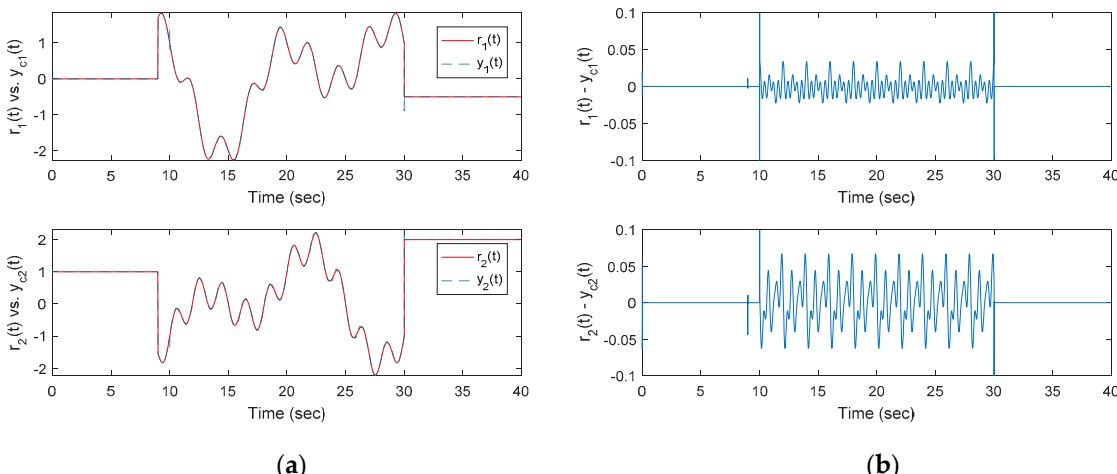

**Figure 8.** (**a**) Time responses of the closed-loop noisy non-square singular system by the optimal OKID-based LQAT without the EID estimator and (**b**) tracking errors of output $e(t) = r(t) - y_c(t)$.

Second, the robustness of the optimal OKID-based LQAT integrated with the modified functional observer-based EID estimator is shown in Figure 9a,b. Additionally, Figure 10 plots the control inputs corresponding to the proposed approach. Notice that $u_c(t) = \eta u_a(t)$ for this non-square system.

A comparison with Figure 8 shows that the tracking errors for the perturbed singular system are significantly reduced.

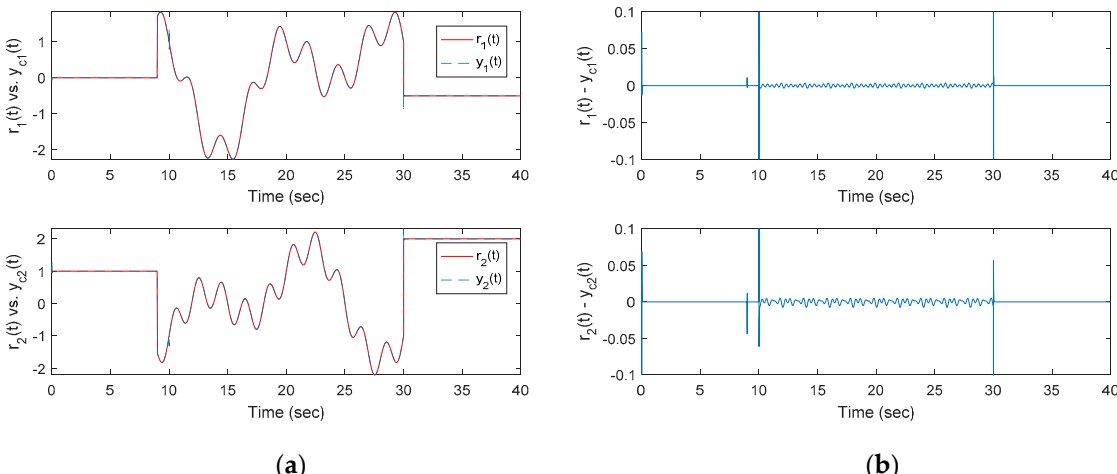

**(a)**                                                      **(b)**

**Figure 9.** (**a**) Time responses of the proposed method and (**b**) errors of output $e(t) = r(t) - y_c(t)$.

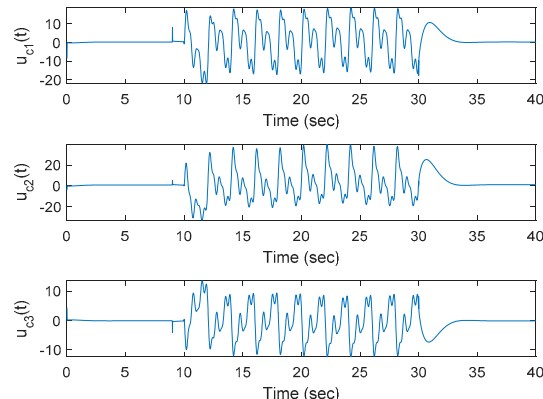

**Figure 10.** Control inputs for the OKID-based LQAT tracker with the proposed EID compensation for the non-square system with mismatched input and output disturbances.

## 7. Conclusions

This paper presents the design of an OKID-based LQAT and the design of a functional observer to estimate the EID for unknown continuous-time square–non-square singular analog systems with all stable zeros and unknown matched–mismatched input and output disturbances. The proposed method significantly improves the tracking performance for systems subject to high-frequency disturbances. Additionally, an optimal OKID-based LQAT integrated with the modified functional observer-based EID estimator is newly proposed for unknown continuous-time singular systems.

The advantages of the proposed method compared to other approaches are as follows: (i) it is capable of rejecting any kind of unknown state-dependent disturbances with a natural frequency smaller than the cut-off frequency; (ii) it does not require a priori information about the disturbances, such as their input matrices at the input or output terminals, rank conditions, differentiation of the measured outputs, or the number of independent signals in the unknown input or output disturbances; (iii) performing the inverse dynamics of the plant is not required; (iv) the system of interest does not need to be of square size; and (v) its implementation is quite simple, since it is a plug-in of an EID estimator that is used to improve any existing controller for a square–non-square servo system.

For communication systems, delays might exist in the measurement outputs or control inputs, since the information is obtained through a communication channel with limited bandwidth. Thus, an extension of the proposed approach with control input constraints for some classes of unknown

time-delay large-scale interconnected linear systems, networked control systems, or non-linear systems will be considered as a future research topic.

**Author Contributions:** Conceptualization and methodology, J.S.-H.T. and T.-J.S.; software and writing—original draft preparation, Y.-F.C. and C.-Y.C.; validation, S.-M.G.; formal analysis, L.-S.S.; writing—review and editing, J.I.C.; funding acquisition, J.S.-H.T. All authors have read and agreed to the published version of the manuscript.

**Funding:** This work was supported by the Ministry of Science and Technology of the Republic of China (MOST 108-2221-E-006-213-MY3 and MOST 107-2221-E-006-203-MY2).

**Conflicts of Interest:** The authors declare no conflict of interest.

## Appendix A. Some Proofs of the Functional Observer-Based EID

*Appendix A.1. Proof of Equivalence Between Condition 1 of Theorem 2 and Condition (31)*

Post-multiplying both sides of Equation (23) by the full-row rank matrix $\begin{bmatrix} H_1 & E_1 & 0 \\ 0 & 0 & I_{n+m} \end{bmatrix}$ yields the left-hand side

$$
\begin{aligned}
\operatorname{rank}\begin{bmatrix} F\overline{A}_\omega & F \\ \overline{C}_\omega\overline{A}_\omega & \overline{C}_\omega \\ \overline{C}_\omega & \overline{C}_\omega & 0 \\ 0 & \overline{E}_\omega \\ F & 0 \end{bmatrix} &= \operatorname{rank}\left(\begin{bmatrix} F\overline{A}_\omega & F \\ \overline{C}_\omega\overline{A}_\omega & \overline{C}_\omega \\ \overline{C}_\omega & 0 \\ 0 & \overline{E}_\omega \\ F & 0 \end{bmatrix}\begin{bmatrix} H_1 & E_1 & 0 \\ 0 & 0 & I_{n+m} \end{bmatrix}\right) \\
&= \operatorname{rank}\begin{bmatrix} F\overline{A}_\omega H_1 & F\overline{A}_\omega E_1 & F \\ \overline{C}_\omega\overline{A}_\omega H_1 & \overline{C}_\omega\overline{A}_\omega E_1 & \overline{C}_\omega \\ \overline{C}_\omega H_1 & \overline{C}_\omega E_1 & 0 \\ 0 & 0 & \overline{E}_\omega \\ I_\kappa & 0 & 0 \end{bmatrix} = \kappa + \operatorname{rank}\begin{bmatrix} F\overline{A}_\omega E_1 & F \\ \overline{C}_\omega\overline{A}_\omega E_1 & \overline{C}_\omega \\ \overline{C}_\omega E_1 & 0 \\ 0 & \overline{E}_\omega \end{bmatrix} = \kappa + \operatorname{rank}\begin{bmatrix} \psi \\ \Delta \end{bmatrix},
\end{aligned}
\tag{A1}
$$

where $FH_1 = FF^+ = I_\kappa$ and $FE_1 = F(I_{n+m} - F^+F) = 0$ have been used therein, along with the right-hand side

$$
\begin{aligned}
\operatorname{rank}\begin{bmatrix} \overline{C}_\omega\overline{A}_\omega & \overline{C}_\omega \\ \overline{C}_\omega & 0 \\ 0 & \overline{E}_\omega \\ F & 0 \end{bmatrix} &= \operatorname{rank}\left(\begin{bmatrix} \overline{C}_\omega\overline{A}_\omega & \overline{C}_\omega \\ \overline{C}_\omega & 0 \\ 0 & \overline{E}_\omega \\ F & 0 \end{bmatrix}\begin{bmatrix} H_1 & E_1 & 0 \\ 0 & 0 & I_{n+m} \end{bmatrix}\right) \\
&= \operatorname{rank}\begin{bmatrix} \overline{C}_\omega\overline{A}_\omega H_1 & \overline{C}_\omega\overline{A}_\omega E_1 & \overline{C}_\omega \\ \overline{C}_\omega H_1 & \overline{C}_\omega E_1 & 0 \\ 0 & 0 & \overline{E}_\omega \\ I_\kappa & 0 & 0 \end{bmatrix} = \kappa + \operatorname{rank}\begin{bmatrix} \overline{C}_\omega\overline{A}_\omega E_1 & \overline{C}_\omega \\ \overline{C}_\omega E_1 & 0 \\ 0 & \overline{E}_\omega \end{bmatrix} = \kappa + \operatorname{rank}(\Delta).
\end{aligned}
\tag{A2}
$$

Equations (A1) and (A2) show that $\operatorname{rank}\begin{bmatrix} F\overline{A}_\omega & F \\ \overline{C}_\omega\overline{A}_\omega & \overline{C}_\omega \\ \overline{C}_\omega & 0 \\ 0 & \overline{E}_\omega \\ F & 0 \end{bmatrix} = \operatorname{rank}\begin{bmatrix} \overline{C}_\omega\overline{A}_\omega & \overline{C}_\omega \\ \overline{C}_\omega & 0 \\ 0 & \overline{E}_\omega \\ F & 0 \end{bmatrix}$.

*Appendix A.2. Proof of Equivalence Between Condition 2 of Theorem 2 and Condition (35)*

Post-multiplying the RHS of Equation (24) by the full-row matrix $\begin{bmatrix} H_1 & E_1 & 0 \\ 0 & 0 & I_{n+m} \end{bmatrix}$ gives

$$\text{rank}\begin{bmatrix} \overline{C}_\omega \overline{A}_\omega & \overline{C}_\omega \\ \overline{C}_\omega & 0 \\ 0 & \overline{E}_\omega \\ F & 0 \end{bmatrix} = \text{rank}\left(\begin{bmatrix} \overline{C}_\omega \overline{A}_\omega & \overline{C}_\omega \\ \overline{C}_\omega & 0 \\ 0 & \overline{E}_\omega \\ F & 0 \end{bmatrix}\begin{bmatrix} H_1 & E_1 & 0 \\ 0 & 0 & I_{n+m} \end{bmatrix}\right) = \kappa + \text{rank}(\Delta). \qquad (A3)$$

Now, the LHS of Equation (24) is the same as

$$\begin{aligned}
&\text{rank}\begin{bmatrix} sF - F\overline{A}_\omega & -F \\ \overline{C}_\omega \overline{A}_\omega & \overline{C}_\omega \\ \overline{C}_\omega & 0 \\ 0 & \overline{E}_\omega \end{bmatrix} = \text{rank}\left(\begin{bmatrix} sF - F\overline{A}_\omega & -F \\ \overline{C}_\omega \overline{A}_\omega & \overline{C}_\omega \\ \overline{C}_\omega & 0 \\ 0 & \overline{E}_\omega \end{bmatrix}\begin{bmatrix} H_1 & E_1 & 0 \\ 0 & 0 & I_{n+m} \end{bmatrix}\right) \\[2mm]
&= \text{rank}\begin{bmatrix} sI_\kappa - F\overline{A}_\omega H_1 & -\psi \\ \widetilde{\Theta} & \Delta \end{bmatrix} = \text{rank}\left(\begin{bmatrix} I_\kappa & \psi\Delta^+ \\ 0 & (I_{n+2p} - \Delta\Delta^+) \\ 0 & \Delta\Delta^+ \end{bmatrix}\begin{bmatrix} sI_\kappa - F\overline{A}_\omega H_1 & -\psi \\ \widetilde{\Theta} & \Delta \end{bmatrix}\right) \qquad (A4) \\[2mm]
&= \text{rank}\begin{bmatrix} sI_\kappa - \Phi & 0 \\ \Omega & 0 \\ \Delta\Delta^+\widetilde{\Theta} & \Delta \end{bmatrix} = \text{rank}\left(\begin{bmatrix} sI_\kappa - \Phi & 0 \\ \Omega & 0 \\ \Delta\Delta^+\widetilde{\Theta} & \Delta \end{bmatrix}\begin{bmatrix} I_\kappa & 0 \\ -\Delta^+\widetilde{\Theta} & I_{2n+2m} \end{bmatrix}\right) \\[2mm]
&= \text{rank}\begin{bmatrix} sI_\kappa - \Phi \\ \Omega \end{bmatrix} + \text{rank}(\Delta), \ \forall s \in C, \ \Re(s) \geq 0.
\end{aligned}$$

It is clear from Equations (A3) and (A4) that Condition 2 of Theorem 2 is equivalent to Equation (35).

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
