# Peer review of "A Modified Functional Observer-Based EID Estimator for Unknown Continuous-Time Singular Systems"

_applsci, doi:10.3390/app10072316_

Round 1
Reviewer 1 Report
The result in this paper, A Modified Functional Observer-Based EID Estimator for Unknown Continuous-Time Singular Systems, appears to be correct and has potential applications in many subjects. Thus, I recommend this paper to be published in Applied Sciences after carry out the following my comments.
1) The abstract and introduction should be polished to highlight the contribution.
2) The linear time-invariant system is considered in this paper. If there are uncertainties in the systems, does the method proposed in this paper still work well? Or, can this method be extended to uncertain linear systems?
3) The application of the proposed study should further clarified.
4) What are the main features of L2 and H-infinity approximation? Explain in detail about the connection and differences among them.
5) The author should clarify the initial conditions?
6) The conclusion part should be further improved.
Author Response
Please see the attachment entitled "Response to Reviewer 1 Comments for applsci-743223R1.pdf".

Reviewer 2 Report
This paper presents a functional observer to estimate the EID for unknown input and output disturbances. The tracking performance of systems subject to high frequency disturbances is much improved. This paper explains advantages of the proposed method over conventional methods.
<Inquiry 1>
The reviewer feels, Sections 2 and 3 does not provide an innovative concept. Thus, to delete or shorten these two chapters seem to be a good idea to concentrate on your proposed concept.
<Inquiry 2>
In pages 6-8, “modified functional observer with unknown input” is explained. Because it is not easy to follow, please provide a summary of the proposed idea in advance in Section 4.
Author Response
Please see the attachment entitled "Response to Reviewer 2 Comments for applsci-743223R1.pdf".

Round 2
Reviewer 1 Report
No comments.